# A novel human pluripotent stem cell gene activation system identifies IGFBP2 as a mediator in the production of haematopoietic progenitors in vitro

**Paolo Petazzi[1], Telma Ventura[2], Francesca Paola Luongo[2], Heather McClafferty[3], Alisha May[2], Helen Alice Taylor[2], Michael J Shipston[3,4], Nicola Romanò[3,4], Lesley M Forrester[2], Pablo Menendez[1,5,6,7,8], Antonella Fidanza[2,9]***

[1]Josep Carreras Leukemia Research Institute, Barcelona, Spain; [2]Centre for Regenerative Medicine, Institute for Regeneration and Repair, University of Edinburgh, Edinburgh, United Kingdom; [3]Centre for Discovery Brain Sciences, Edinburgh Medical School, Biomedical Sciences, University of Edinburgh, Edinburgh, United Kingdom; [4]Zhejiang University-University of Edinburgh Joint Institute, Zhejiang University School of Medicine, Zhejiang University, Haining, China; [5]CIBER-ONC, ISCIII, Barcelona, Spain; [6]Institució Catalana de Recerca i Estudis Avançats (ICREA), Barcelona, Spain; [7]Department of Biomedicine, School of Medicine, University of Barcelona, Barcelona, Spain; [8]Pediatric Cancer Centre Barcelona-Institut de Recerca Sant Joan de Deu (PCCB-SJD), Barcelona, Spain, Barcelona, Spain; [9]Edinburgh Medical School, Biomedical Sciences, University of Edinburgh, Edinburgh, United Kingdom

*For correspondence:
afidanza@ed.ac.uk

**Competing interest:** The authors declare that no competing interests exist.

## eLife Assessment

This study presents **useful** findings to inform and improve the in vitro differentiation of hematopoietic progenitor cells from human induced pluripotent stem cells. Relying on a well-characterised technical approach, the data analysis is overall **solid** and reasonably supports the main conclusions.

**Abstract** A major challenge in the stem cell biology field is the ability to produce fully functional cells from induced pluripotent stem cells (iPSCs) that are a valuable resource for cell therapy, drug screening, and disease modelling. Here, we developed a novel inducible CRISPR-mediated activation strategy (iCRISPRa) to drive the expression of multiple endogenous transcription factors (TFs) important for in vitro cell fate and differentiation of iPSCs to haematopoietic progenitor cells. This work has identified a key role for IGFBP2 in developing haematopoietic progenitors. We first identified nine candidate TFs that we predicted to be involved in blood cell emergence during development, then generated tagged gRNAs directed to the transcriptional start site of these TFs that could also be detected during single-cell RNA sequencing (scRNAseq). iCRISPRa activation of these endogenous TFs resulted in a significant expansion of arterial-fated endothelial cells expressing high levels of IGFBP2, and our analysis indicated that IGFBP2 is involved in the remodelling of metabolic activity during in vitro endothelial to haematopoietic transition. As well as providing fundamental new insights into the mechanisms of haematopoietic differentiation, the broader applicability of iCRISPRa provides a valuable tool for studying dynamic processes in development and for recapitulating abnormal phenotypes characterised by ectopic activation of specific endogenous gene expression in a wide range of systems.

## Introduction

The haematopoietic system develops early during gestation through the so-called 'waves' of haematopoietic progenitors that arise in different anatomical regions and result in the production of various progenitor and stem cells (*Medvinsky and Dzierzak, 1996*; *Palis et al., 1999*; *Patel et al., 2022*; *Böiers et al., 2013*; *Hoeffel et al., 2015*). The precise signalling pathways leading to the production of haematopoietic stem and progenitor cells (HSPCs) during embryonic development are yet to be completely defined, posing a limitation on how to recapitulate the process in vitro from pluripotent stem cells.

During development, HSPCs are generated by a subset of endothelial cells, known as haemogenic endothelium (*Jaffredo et al., 1998*; *Zovein et al., 2008*; *Bertrand et al., 2010*; *Boisset et al., 2010*), via the endothelial to haematopoietic transition (EHT) *Ottersbach, 2019*. During the EHT, endothelial cells undergo profound transcriptional remodelling whereby the expression of endothelial genes is gradually downregulated, and the transcription of the haematopoietic program is initiated (*Swiers et al., 2013*) while the cells round up and eventually detach to enter the circulation (*Kissa and Herbomel, 2010*; *Eilken et al., 2009*).

To explore the molecular control on the development of the haematopoietic system and to address the differences with the in vitro differentiation of induced pluripotent stem cells (iPSCs), we compared our single-cell transcriptomics analysis of in vitro-derived haemogenic endothelium and early progenitors (*Fidanza et al., 2020*) to that of the in vivo HSC-primed human haemogenic endothelium (*Zeng et al., 2019*). We then developed a novel doxycycline (DOX)-inducible CRISPR gene activation system to assess the role of the genes that were expressed at a lower level within in vitro-derived cells compared to their in vivo counterparts. We employed single-cell RNA sequencing (scRNAseq) to track the presence of guide RNAs and we monitored the phenotypic effects of gene activation. With this experimental pipeline, we identified and functionally validated a novel role for IGFBP2, IGF binding protein 2, in the development of in vitro progenitors and showed that IGBP2 remodels the metabolic activity during in vitro EHT.

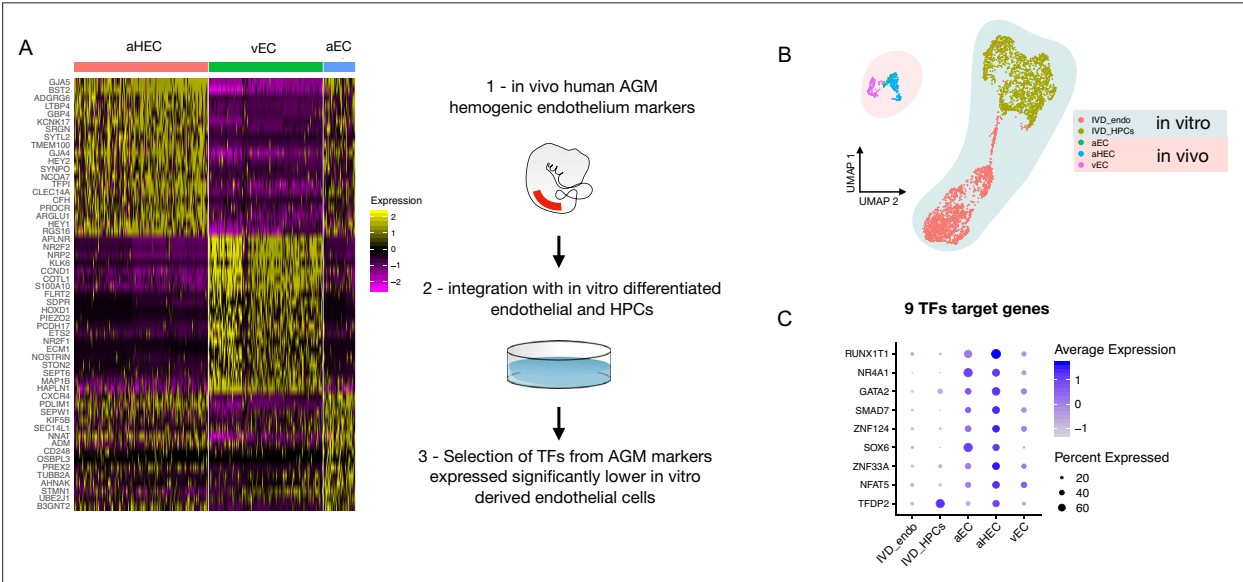

**Figure 1.** Comparison of in vitro induced pluripotent stem cell (iPSC)-derived and in vivo aorta-gonad-mesonephros (AGM)-derived endothelial cells identifies nine differentially expressed transcription factors. (**A**) Schematic of the analytic pipeline used to identify the target genes. (**B**) Integrative analysis of single-cell transcriptome of in vitro-derived endothelial (IVD_Endo) and haematopoietic cells (IVD_HPCs) with in vivo developed endothelial cells (venous, vEC; arterial, aEC; arterial haemogenic, HECs) from human embryos (CS12-CS14) visualised on UMAP dimensions. (**C**) Target genes expression level showing higher expression in arterial haemogenic endothelium in vivo than in vitro-derived cells.

The online version of this article includes the following figure supplement(s) for figure 1:

**Figure supplement 1.** Gene expression profile of the target genes in an additional human AGM dataset.

## Results

### Comparison of iPSC-derived endothelial cells with AGM endothelial cells dataset identifies nine differentially expressed transcription factors

We, and others, have shown that in vitro differentiation of human iPSCs (hiPSCs) is a powerful tool to model intraembryonic haematopoiesis (*Fidanza et al., 2020*; *Ng et al., 2016*; *Calvanese et al., 2022*; *Sturgeon et al., 2014*). To understand the molecular basis underlying the challenges associated with the in vitro production of blood stem and progenitor cells in vitro from differentiating iPSCs, we compared our scRNAseq dataset from differentiating iPSCs (*Fidanza et al., 2020*) to that of cells derived in vivo from the human aorta-gonad-mesonephros (AGM) region (*Zeng et al., 2019*). We integrated the transcriptomic data of in vitro-derived endothelial (IVD_Endo) and haematopoietic cells (IVD_HPC) with that of arterial endothelial cells (aEC), arterial haemogenic cells (aHEC), and venous endothelial cells (vEC) derived from human embryos collected between Carnegie stages 12 and 14 (*Figure 1A and B*). To identify possible target genes that could be manipulated in vitro to improve iPSCs differentiation, we first determined the transcription factors marking the aHEC in vivo (*Supplementary file 1*). Then, we filtered the transcription factor genes that were expressed at lower levels in the IVD_Endo and IVD_HPC. This strategy identified nine target transcription factors *RUNX1T1*, *NR4A1*, *GATA2*, *SMAD7*, *ZNF124*, *SOX6*, *ZNF33A*, *NFAT5*, and *TFDP2* (*Figure 1C*), whose expression was detected across other AGM datasets in the haemogenic endothelium (*Calvanese et al., 2022*; *Figure 1—figure supplement 1*).

### Development of a DOX-inducible dCAS9-SAM activation system in hiPSCs

We previously developed an all-in-one Synergistic Activator Mediator system, UniSAM, that mediates the transcriptional activation of endogenous gene expression (*Fidanza et al., 2017*). To activate the nine target genes identified in this study, we developed a novel DOX-inducible SAM (iSAM) cassette targeted into the *AAVS1* locus of human iPSCs (*Figure 2A*). We demonstrated this strategy could activate *RUNX1C* expression in HeLa cells and that this was correlated with the DOX concentration in a linear manner (*Figure 2—figure supplement 1A and B*). To verify gene activation in human PSCs at single-cell resolution, we employed a RUNX1C-GFP human embryonic stem cell (hESC) reporter cell line (*Figure 2—figure supplement 1C–G*). As predicted, the level of expression of the mCherry tag, the fluorescent mCherry reporter tag within the iSAM cassette, was proportional to the concentration of DOX (*Figure 2—figure supplement 1D and E*) and to the number of cells in which RUNX1C-GFP was activated (*Figure 2—figure supplement 1D–F*). Furthermore, the RUNX1C expression level, measured by the mean fluorescence intensity of the RUNX1C-GFP reporter, also correlated with the concentration of DOX (*Figure 2—figure supplement 1G*). We then tested the iSAM cassette in the iPSC line (SFCi55) (*Figure 2B–D*). Only when iPSCs were transfected with both iSAM and the gRNA directed to *RUNX1C* and treated with DOX, was the expression of the *RUNX1C* gene and RUNX1 protein detected (*Figure 2—figure supplement 1B and C*).

We then targeted the iSAM cassette into the *AAVS1* locus using a zinc finger nuclease (ZFN) strategy (*Fidanza et al., 2020*; *Yang et al., 2017*; *Lopez-Yrigoyen et al., 2018*). iPSC clones that had specifically integrated the iSAM cassette into the *AAVS1* locus were validated by genomic PCR screening (*Figure 2—figure supplement 1A and B*) and sequencing. The *AAVS1* locus has been reported to be a 'safe harbour' site that is resistant to epigenetic silencing and indeed, we had previously demonstrated that transgenes inserted into the *AAVS1* locus under the control of the constitutively active CAG promoter were efficiently expressed both in undifferentiated and in differentiated iPSCs (*Yang et al., 2017*; *Lopez-Yrigoyen et al., 2019*; *Lopez-Yrigoyen et al., 2018*; *May et al., 2023*). However, after the iSAM line had been established and cultured under self-renewal conditions, we noted a dramatic reduction in the number of mCherry+ cells in undifferentiated iSAM iPSCs upon DOX induction. We predicted this to be due to transgene silencing of the rTTA DOX-inducible cassette (*Figure 2—figure supplement 2C*). To overcome this problem, we treated the iSAM iPSC line with an inhibitor of histone deacetylases (HDACs), sodium butyrate (SB), reported to have no adverse effect on iPSC maintenance (*Kang et al., 2014*; *Zhang et al., 2014*) and we also confirmed that the treatment with SB had no effect on viability and cell proliferation in our culture conditions

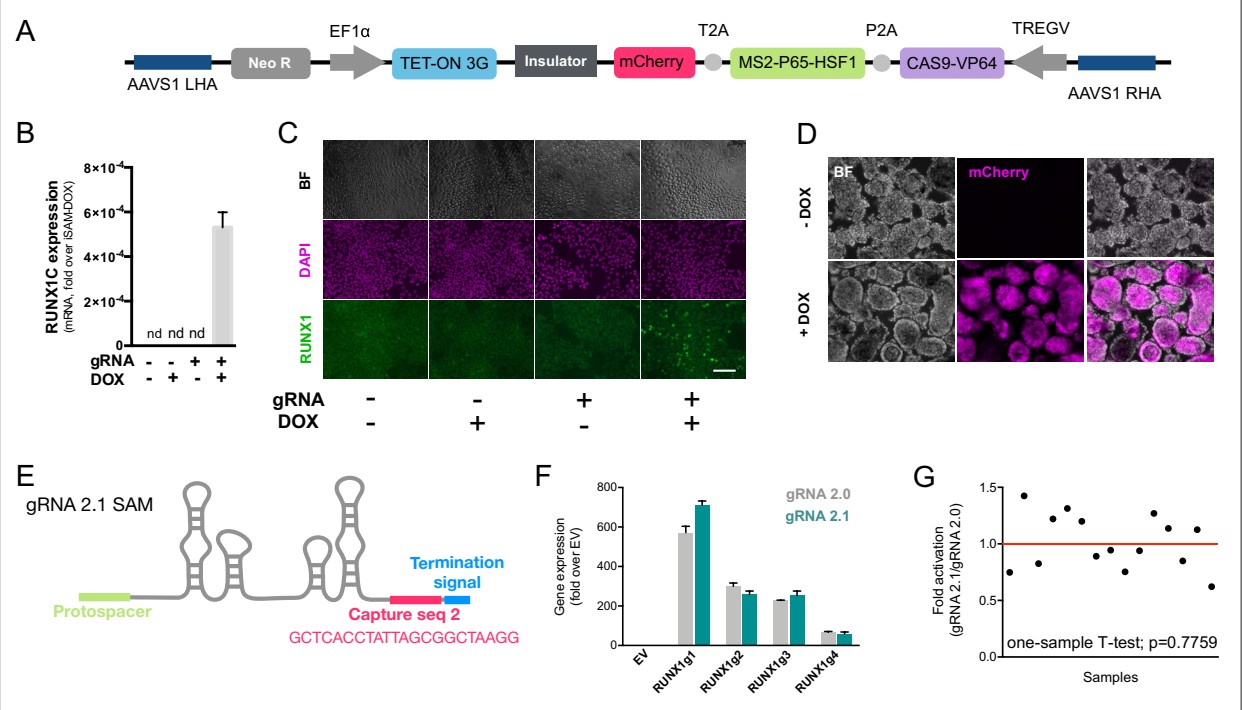

**Figure 2.** The inducible SAM (iSAM) cassette successfully mediates activation of endogenous gene expression upon doxycycline (DOX) induction. (**A**) Schematic of the iSAM cassette containing the TET-on system under the control of EF1α and dCAS9-P2A-MS2-p65-HSF1-T2A-mCherry under the rTTA responsive elements, separated by genetic silencer and flanked by AAVS1 specific homology arms. (**B**) *RUNX1C* gene expression activation after transient transfection of the iSAM plasmid and gRNAs in presence or absence of DOX in human induced pluripotent stem cell (iPSC) line (n=3 from independent transfections). (**C**) RUNX1 protein expression upon iSAM activation after transient transfection of the iSAM plasmid and gRNAs with DOX in human iPSC line detected by immunostaining. (**D**) Expression of the iSAM cassette reported by mCherry tag during the differentiation protocol, the representative images (bright field - BF, and fluorescence) show embryoid bodies at day 3 of differentiation. (**E**) Schematic of the gRNA 2.1 containing the capture sequence for detection during the single-cell RNA sequencing (scRNAseq) pipeline. (**F**) *RUNX1C* gene activation level obtained using either the gRNA 2.0 or 2.1 backbone (n=3 from independent transfections of the 4 different gRNAs). (**G**) Statistical analysis of the gRNAs activation level showing no significant variation following addition of the capture sequence (n=3 for each of the 4 different gRNAs).

The online version of this article includes the following source data and figure supplement(s) for figure 2:

**Source data 1.** Spreadsheet source file containing the source data used for the plots in *Figure 2*.

**Figure supplement 1.** Functional validation of the iSAM plasmid.

**Figure supplement 1—source data 1.** Spreadsheet source file containing the source data used for the plots in *Figure 2—figure supplement 1*.

**Figure supplement 2.** Validation of the iSAM hiPSC cell lines.

**Figure supplement 2—source data 1.** Spreadsheet source file contains the source data used for the plots in *Figure 2—figure supplement 2*.

**Figure supplement 2—source data 2.** Raw uncropped images of the entire gels used in *Figure 2—figure supplement 2*.

**Figure supplement 2—source data 3.** Raw uncropped images of the entire gels used in *Figure 2* figure with labelling of the band highlighted in *Figure 2—figure supplement 2*.

(*Figure 2—figure supplement 2F and G*). A short 48 hr treatment significantly increased the number of mCherry+ cells upon DOX induction, proportional to the SB concentration (*Figure 2—figure supplement 2D and E*). We, therefore, maintained the iSAM iPSCs in the presence of SB and this fully restored the inducibility of the transgene with virtually all cells expressing mCherry in the presence of DOX (*Figure 2—figure supplement 2G*). We did not treat the cells with SB during the differentiation since we detected robust expression of the iSAM cassette during the differentiation (*Figure 3—figure supplement 1F*). Furthermore, since the HDACs are also involved in the chromatin remodelling during the EHT process (*Thambyrajah et al., 2018*), we predicted that SB treatment would negatively affect the differentiation.

To test the effect of activating the nine target genes on the transcriptomes of differentiating iPSC cells, we engineered the gRNAs to allow their detection within the scRNAseq pipeline (*Replogle*

*et al., 2020*). We inserted a capture sequence prior to the termination signal to avoid any alteration in the secondary structure of the loops thus preserving the binding of the synergistic activators of the SAM system to the stem loops of the gRNAs. Of the two capture sequences available (*Replogle et al., 2020*), we decided to use the one that was predicted to result in fewer secondary structure alterations and this new gRNA was named 2.1 (*Figure 2E*). We compared the activation level achieved with the new 2.1 gRNA to that of the original 2.0 backbone using various gRNAs targeting *RUNX1C* (*Figure 2F*). These results convincingly demonstrated that the addition of the capture sequence in the gRNA 2.1 does not alter the level of endogenous gene activation that could be achieved (*Figure 2G*). Altogether, these results show that the iSAM system is able to induce gene expression that predictably translates into an increased protein expression and thus provides a platform to steer phenotypical changes in cell identity.

To activate our target genes, we designed 5–7 gRNAs in the 200 bp upstream of the transcriptional start sites of each of the 9 target genes. We subcloned a total of 49 gRNAs (*Supplementary file 1*) into the gRNA 2.1 backbone and packaged them into lentiviral particles (herein referred to as the AGM library) as well as a non-targeting (NT) gRNA that was used as control. The iSAM iPSC line was transduced with the targeting gRNAs or the control NT gRNA to generate the iSAM_AGM and iSAM_NT iPSCs line, respectively (*Figure 3—figure supplement 1A*). After puromycin selection, their

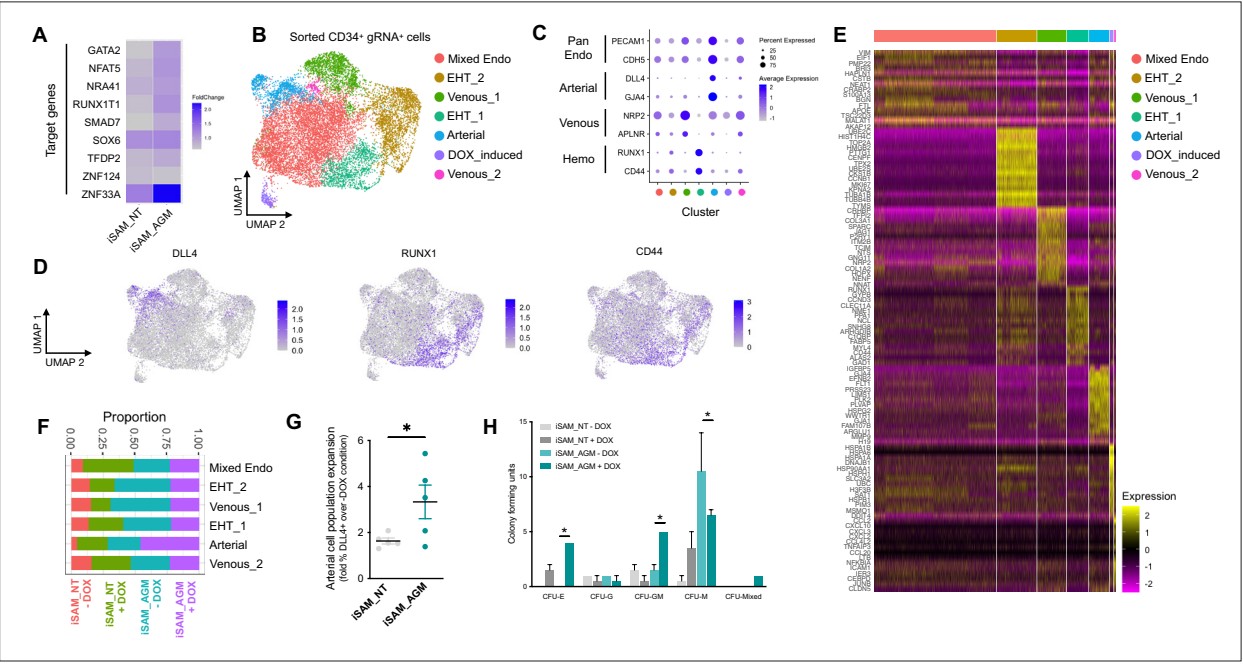

**Figure 3.** Single-cell RNA sequencing (scRNAseq) in combination with CRISPR activation identifies arterial cell type and functional haematopoietic expansion in association with activation of the nine target genes. (**A**) Gene expression profile of target genes following target genes' activation, heatmap shows the expression level of the target genes in the iSAM_NT and iSAM_AGM treated with doxycycline (DOX) following normalisation on the -DOX control. (**B**) Dimension reduction and clustering analysis of the scRNAseq data following activation, filtered on cells where the gRNA expression was detected. (**C**) Arterial (*GJA4*, *DLL4*), venous (*NRP2*, *APLNR*), and haemogenic marker (*CD44*, *RUNX1*) expression distribution in the clusters indicated by the colour. (**D**) Expression distribution visualised on the UMAP plot showing the location of arterial cells marked by *DLL4*, and haemogenic endothelium marked by *CD44* and *RUNX1*. (**E**) Heatmap of the top 15 marker genes for each of the clusters. (**F**) Contribution of the different libraries to the clusters showing that arterial cell cluster is overrepresented in the iSAM_AGM treated with DOX, compared to the other libraries. (**G**) Expansion of the arterial population assessed by the membrane marker expression of DLL4+ following targets' activation, quantified by flow cytometry at day 8 of differentiation. (Data are normalised on the iSAM_NT+DOX sample, n=5 independent differentiations, *p=0.0417 paired t-test.) (**H**) Colony-forming potential of the suspension progenitor cells derived from the two lines treated with or without DOX following OP9 coculture activation, data show the colony obtained for 104 CD34+ input equivalent (n=3 from independent differentiations *p<0.05, Tukey's two-way ANOVA).

The online version of this article includes the following source data and figure supplement(s) for figure 3:

**Source data 1.** Spreadsheet source file containing the source data used for the plots in *Figure 3*.

**Figure supplement 1.** Experimental design and further characterisation of the activation results.

**Figure supplement 1—source data 1.** Spreadsheet source file containing the source data used for the plots in *Figure 3—figure supplement 1*.

integration in the genome was also confirmed by PCR and sequencing (*Figure 2—figure supplement 2H and I*).

## scRNAseq in combination with CRISPR activation identifies arterial cell-type expansion in association with higher haematopoietic progenitor potential

To assess the transcriptional changes in response to the activation of the target genes, we differentiated the iSAM iPSCs, induced with DOX and subjected them to scRNAseq using the 10× pipeline. After 10 days of differentiation, in the presence or absence of DOX, we FAC-sorted live CD34+ cells from iSAM_AGM and the iSAM_NT iPSCs (*Figure 3—figure supplement 1B*). Following data filtering, we selected cells in which the gRNAs expression was detected and showed that our approach activated all the target genes, except for ZFN124. A higher level of expression of *RUNX1T1*, *NR4A1*, *GATA2*, *SMAD7*, *SOX6*, *ZNF33A*, *NFAT5*, *TFDP2* was observed following DOX treatment of cells of iSAM_AGM cells compared to the iSAM_NT cells (*Figure 3A*, *Figure 2—figure supplement 2G*). To study the effect of gene activation on transcriptional and cellular phenotype, we performed clustering analysis and detected a total of seven clusters (*Figure 3B–E*). A high level of *GJA4* and *DLL4* expression was used to annotate the arterial-like cell cluster, while high levels of haemogenic markers, such as *RUNX1* and *CD44*, were used to annotate the haemogenic clusters, EHT_1 and EHT_2 (*Figure 3B, C and D*). All the clusters expressed pan-endothelial markers such as *PECAM1* and *CDH5*, coding for CD31 and VECAD, respectively, which get progressively downregulated in EHT_1 and EHT_2, as expected for cells undergoing the EHT process. To assess the effect of the activation on cell identity, we analysed the proportion of the cell clusters between the libraries and detected a significant expansion of the arterial cluster in the DOX-induced iSAM-AGM compared to the iSAM_NT cells (*Figure 3E*, *Figure 3—figure supplement 1D and I*). To validate the effect of the activation on the expansion of arterial cell population, we analysed their prevalence by flow cytometry. Although DOX treatment resulted in an average 1.63±0.13-fold increase of CD34$^+$DLL4$^+$ cells in the control iSAM_NT sample (*Figure 2—figure supplement 2A*), the increase observed in the iSAM-AGM cells was significantly greater, with an expansion of 3.33±0.73-fold increase of CD34$^+$DLL4$^+$ immunophenotypic arterial cells identified by flow cytometry (*Figure 3F*). To assess the effect of the different cell composition in the CD34+ compartment upon activation on the emergence of colony-forming progenitors, we isolated CD34+ cells using magnetic beads and cocultured 20,000 cells on OP9 supportive stromal cells for 7 days in the presence of haematopoietic differentiation cytokines. After 1 week, the progenitor cells were assessed by colony-forming unit (CFU) assays and scored 14 days later. We detected an increased number of CFU-E (colony-forming unit erythroid) and CFU-GM (colony-forming unit granulocyte/macrophage) and a reduction of CFU-M (colony-forming unit macrophage) in iSAM-AGM samples cultured in the presence of DOX compared to the absence of DOX but no significant effect of DOX in SAM_NT samples (*Figure 3G*). These data indicate that activation of target transcription factors using our novel CRISPR strategy results in transcriptional remodelling and a steer in cell identity that we detected as a functional difference in the haematopoietic progenitor profile.

## The addition of IGFBP2 to the in vitro differentiation leads to a higher number of functional haematopoietic progenitor cells

To better understand the molecular mechanism behind the increased progenitor development, we compared the expression profile of the arterial cells between the different activation libraries, and we obtained a list of genes upregulated upon activation of the targets (*Supplementary file 1*). The most upregulated gene, *IGFBP2*, was expressed at significantly higher levels in the iSAM_AGM library in the presence of DOX compared to the others (*Figure 4A*, *Supplementary file 1*). Although the arterial cells expressed *IGFBP2* at the highest level compared to other cell types, the activation of the gene was not cell-type specific and it was detected across the various clusters (*Figure 3—figure supplement 1H*). We then compared the gRNA distribution in these arterial cells from the iSAM_AGM treated with DOX to that of arterial cells without activation. We observed a significant enrichment of the *RUNX1T1-specific* gRNAs (*Supplementary file 1*), indicating that the increased *IGFBP2* expression could be downstream of *RUNX1T1* activation. IGF binding protein 2 is a member of the family of IGF binding proteins and is thought to be secreted from cells where it then binds IGF1, IGF2, and other extracellular matrix proteins, modulating their function. IGF1 and IGF2 are commonly used in

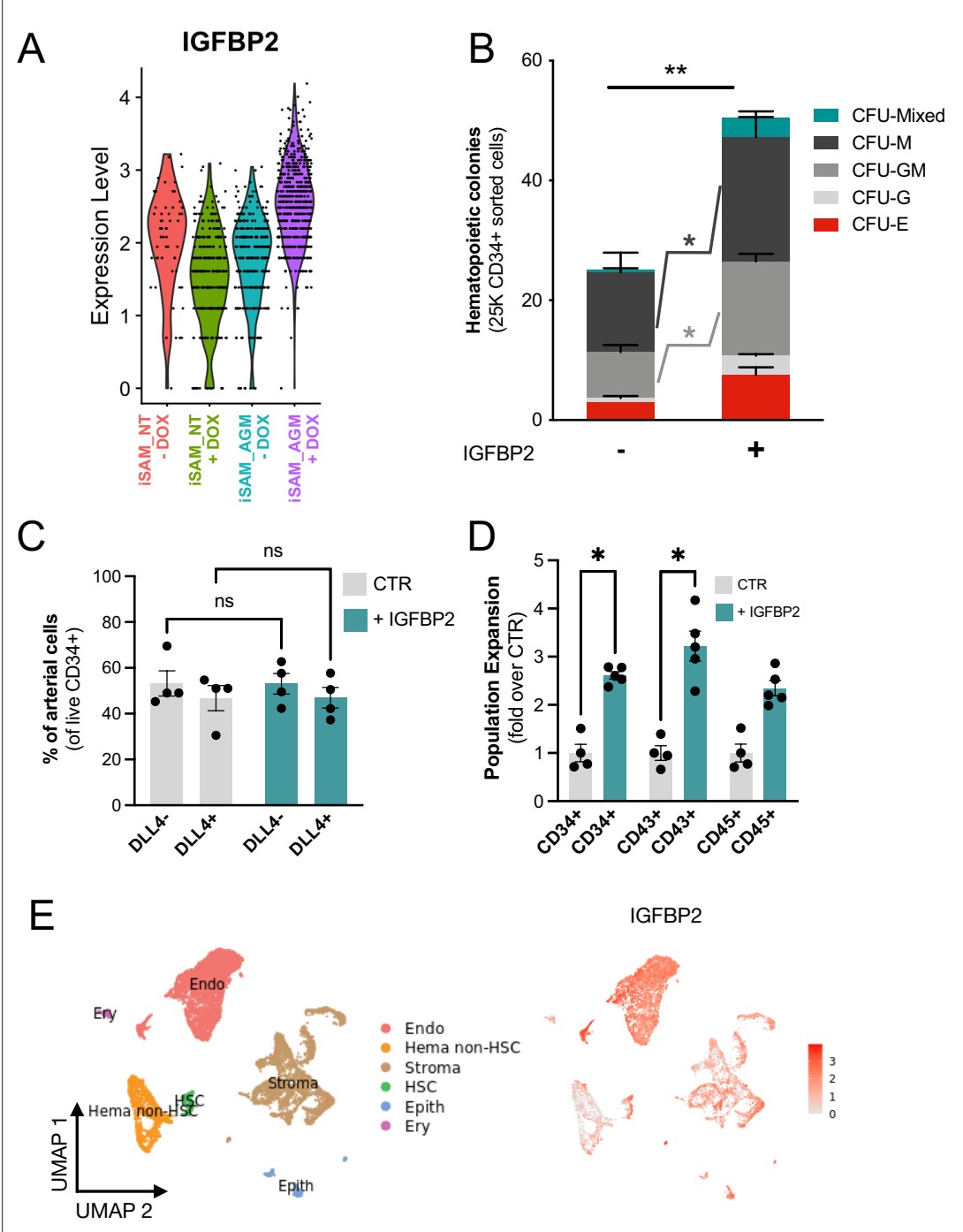

**Figure 4.** IGFBP2 addition to the in vitro differentiation leads to a higher number of functional haematopoietic progenitor cells. (**A**) Violin plot of *IGFBP2* expression profile in the arterial cells obtained from the different conditions, in the presence or absence of gRNAs and doxycycline (DOX). (**B**) Number of haematopoietic colonies obtained after coculture on OP9 in the presence or absence of IGFBP2 (n=3–4 from independent differentiations, **p=0.0080, Sidak's two-way ANOVA). (**C**) Percentage of DLL4+ arterial cells differentiation within the CD34+ compartment analysed by flow cytometry in day 8 embryoid bodies (EBs) (n=4 from independent differentiations, two-way ANOVA, ns = p>0.99). (**D**) Expansion of haematopoietic progenitors analysed using markers' expression on suspension progenitors derived after coculture of CD34+ cells onto OP9 support (data are expressed as fold over the CTR in the absence of IGFBP2; n=4 from independent differentiations, *p<0.02, Sidak's two-way ANOVA). (**E**) Single-cell transcriptomic analysis of developing aorta-gonad-mesonephros (AGM) collected from human embryos at Carnegie stages14 and 15 enriched for CD31+ and CD34+ showing the *IGFBP2* expression profile in vivo in the AGM.

The online version of this article includes the following source data for figure 4:

**Source data 1.** Spreadsheet source file containing the source data used for the plots in *Figure 4*.

differentiation protocols (*Ditadi et al., 2015*), including ours (*Ventura et al., 2023*; *Fidanza et al., 2020*), due to their direct role in blood development. To test if the increased frequency of functional haematopoietic progenitors was due to IGFBP2 signalling, presumably derived from the arterial cells, we supplemented the media with IGFBP2 at 100 ng/ml after the induction of endothelial cell differentiation. To explore the role of IGFBP2, we employed the parental iPSCs line, SFCi55, from which the iSAM line was derived. We isolated CD34+ cells at day 8 and cocultured them on OP9 cells in the presence of IGFBP2 for 1 week, then tested for their haematopoietic clonogenic potential using CFU assays (*Figure 4B*). Cells treated with IGFBP2 showed a significant increase in the total number of haematopoietic CFU colonies compared to cells in control cultures. To assess whether IGFBP2 also affected the production of arterial cells themselves, we analysed the proportion of DLL4+ cells, but no difference was detected in the presence of IGFBP2 (*Figure 4C*). This implies that the mechanism of action of IGFBP2 is different from that mediated by the gene activation that led to an expansion of the arterial population. To further understand the relationship between colonies' potential with the arterial identity, we sorted CD34+ into DLL4+ and DLL4- and cultured them on OP9 coculture for a week prior to methylcellulose assay. CD34+DLL4+ and CD34+DLL4- plated on OP9 showed different capacity to generate suspension cells (*Figure 5—figure supplement 1D*), in line with the colonies' formation results showing that the DLL4- contains the largest progenitor activity. This aligns with the observation that the EHT process coincides with the downregulation of arterial markers such as *DLL4* and the upregulation of haemogenic markers such as *RUNX1* (*Figure 3C*).

This observation supports our hypothesis that the change in haematopoietic progenitor production following activation in the iSAM_AGM line is associated with both differential gene expression within the arterial cells rather and their expansion. We then focused on the characterisation of the cells derived from the CD34+ cells after coculture with the OP9 in the presence of IGFBP2. Our results showed a significant expansion of the CD34+ and CD43+ cell populations, further supporting our hypothesis (*Figure 4D*). To address the potential role of *IGFBP2* in vivo, we analysed single-cell sequencing data from the human AGM at Carnegie stages 14 and 15. *IGFBP2* is highly expressed within the AGM niche by stromal and epithelial cells and, most importantly, highest in endothelial cells (*Figure 4E*). Together these data show that IGFBP2 addition results in increased haematopoietic blood production in vitro and that the endothelial compartment is the most likely source of IGFBP2 both in vitro and in vivo.

## IGFBP2 enhances metabolic dependency on oxidative phosphorylation of differentiating endothelial cells

Following the observation that IGFBP2 supports haematopoietic progenitor differentiation in vitro from hiPSCs, we performed a time course single-cell RNA experiment of SFCi55 iPSCs differentiated in its presence. We isolated CD34+ cells and plated them in EHT culture on laminin (*Ventura et al., 2023*), rather than on OP9 support, to assess exclusively the specific effects of IGFBP2- on differentiating iPSCs. FAC-sorted single/live adherent cells from day 10 and day 13 in the presence and absence of IGFBP2 were subjected to scRNAseq (*Figure 3—figure supplement 1C*). Our time course transcriptomic analyses showed that IGFBP2 induced a change in the transcriptome, specifically on day 13 (*Figure 5A*). A cluster of endothelial cells enriched for genes associated with the KEGG pathway of growth factor binding was detected almost exclusively at day 13 (*Figure 5—figure supplement 1F*); these cells displayed a different transcriptional signature (as indicated by a shift in their position in the UMAP embedding) in the presence of IGFBP2, and were enriched upon treatment (*Figure 5—figure supplement 1F*). Interestingly, this cluster showed expression of *GJA4* across the cells, indicating their broad arterial identity, with a specific reduction of *DLL4* expression level concomitantly to an upregulation of *RUNX1* in the region of the cluster induced by IGFBP2 (*Figure 5B*). This increase in haemogenic identity within the endothelial cell compartment is consistent with the observation of increased functional progenitor production. This suggests that IGFBP2 supports arterial cells in the acquisition of haemogenic capacity.

We compared the transcriptome of cells at day 13 in the presence and absence of IGFBP2 and, using KEGG enrichment analysis, we observed that the genes that were upregulated by IGFBP2 were highly enriched in the oxidative phosphorylation term (*Figure 5C and D*). Since the metabolic switch between glycolytic to mitochondrial metabolism has been previously reported in definitive haematopoiesis (*Azzoni et al., 2021*; *Oburoglu et al., 2022*), we tested whether the addition of IGFBP2 could

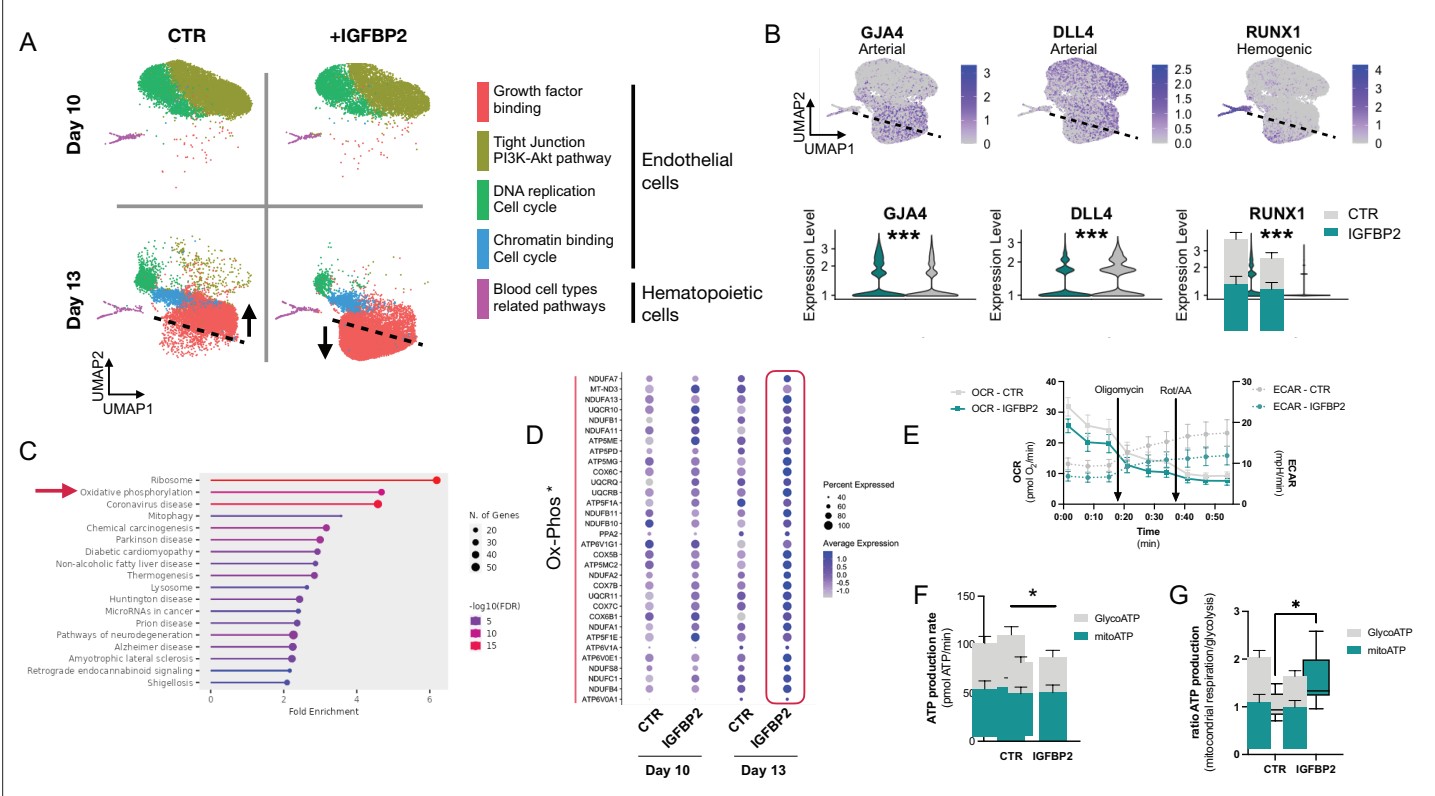

**Figure 5.** IGFBP2 alters cell metabolism by inducing a reduction in glycolytic ATP production. (**A**) Clustering analysis of the single-cell transcriptomic time course analysis of differentiating cells at day 10 and day 13 in the absence (CTR) or presence of IGFBP2. Arrows indicate the difference in the clustering due to the addition of IGFBP2 compared to control. (**B**) Expression profile of arterial markers, *GJA4* and *DLL4*, and haemogenic marker *RUNX1* (top - the dashed line shows the location of the shift in gene expression of cells treated with IGFBP2) and their expression profile in the endothelial cells cluster marked by growth factor binding in the absence (CTR) and in the presence of IGFBP2 (*GJA4* p=1E$^{-54}$, *DLL4* p=1.2E$^{-119}$, *RUNX1* p=8.2E$^{-163}$). (**C**) KEGG enrichment analysis of the genes upregulated at day 13 upon IGFBP2 treatment. The arrow shows the ranking of the oxidative phosphorylation pathway. (**D**) Dot plot showing the expression profile of the genes coding for the enzyme of the oxidative phosphorylation pathway. (**E**) Oxygen consumption rate (OCR) and extracellular acidification rate (ECAR) profile in cells at day 13 of differentiation reporting mitochondrial respiration and glycolysis, respectively. (**F**) ATP production rate divided by that deriving from glycolysis and from mitochondrial respiration, in cells treated with IGFBP2 and controls at day 13. (**G**) Ratio of the ATP production between glycolysis and mitochondrial respiration in cells treated with IGFBP2 and controls at day 13.

The online version of this article includes the following source data and figure supplement(s) for figure 5:

**Source data 1.** Spreadsheet source file containing the source data used for the plots in *Figure 5*.

**Figure supplement 1.** Additional analyses of the effect of IGFBP2 addition during in vitro differentiation.

**Figure supplement 1—source data 1.** Spreadsheet source file containing the source data used for the plots in *Figure 5—figure supplement 1*.

result in a different ATP production profile. We analysed the ATP production at day 13, when RUNX1 expression was induced by IGFBP2 addition, by using specific inhibitors of the complex I and II and complex V to quantify the intracellular ATP and mitochondrial synthesis, respectively. We detected a reduction in glycolytic-derived ATP (*Figure 5E and F*), which translates to a higher contribution of mitochondria metabolism for IGFBP2-treated cells (*Figure 5G*). We then looked at the expression levels of genes encoding the enzymes of the glycolytic pathway and its checkpoints, glucose and lactate transporters, monocarboxylate transporters, and the enzymes associated with hexokinase and phosphofructose reactions. We observed a general downregulation of the glycolytic enzymes and its checkpoints, with the exception of *PFKM* and *SLC16A1* (*Figure 5—figure supplement 1A and B*), which were expressed at lower levels in the IGFBP2-treated cells at day 13 compared to the control on the same day. These differences were not observed on day 10 cells, which is consistent with our previous observations that the IGFBP2-induced remodelling of the endothelial cells' transcriptome happens exclusively on day 13.

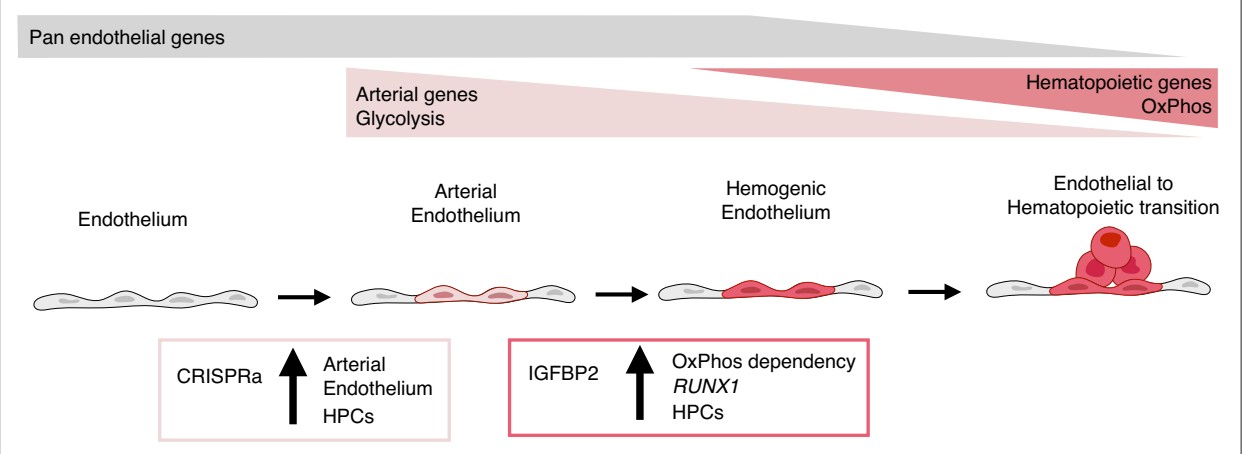

**Figure 6.** Model summarising the results. During development, some endothelial cells undergo arterialisation, as identified by their arterial genes' expression profile (e.g. *DLL4*). Arterial cells, characterised by high dependency on glycolysis, are expanded by our CRISPR activation approach, resulting in more blood production since these cells are the cell-of-origin of the haemogenic endothelium. Once arterial cells commit to haemogenic endothelium fate, they start to express haematopoietic genes (e.g. *RUNX1*); this process is enhanced by IGFBP2 via induction of both *RUNX1* expression and increased dependency on oxidative phosphorylation, known to be important for the progression of the endothelial to haematopoietic transition.

To assess whether the higher clonogenic potential and effect on the bioenergetic profile of the cells treated with IGFBP2 was the result of an increase in proliferation, we analysed the cell cycle profile of suspension haematopoietic cells obtained from the OP9 cocultured in the presence of IGFBP2. No differences in the cell cycle distribution of haematopoietic progenitors (*Figure 3—figure supplement 1E*) was observed, indicating that the detected increase in haematopoietic progenitors is not likely to be a consequence of their increased cycling. We, therefore, concluded the effect of IGBP2 occurs prior to, or during their emergence via the EHT process.

In summary, our results show that the activation of the nine target transcription factors, expressed at higher levels in vivo within the arterial cells in the AGM region, leads to the expansion of the arterial cells and, consequently, to an increase in progenitor activity (*Figure 6*). IGFBP2 was identified as the most upregulated gene in the arterial cell upon gene activation. The addition of IGFBP2 to the differentiation culture induced the upregulation of *RUNX1* and OxPhos genes and accelerated the metabolic dependency on OXPHOS in association with improved progenitor activity (*Figure 6*).

## Discussion

The complexity and dynamism of developmental haematopoiesis in vivo have imposed challenges in accurately reproducing the process in vitro. We hypothesised that this could, in part, be due to the inability to recapitulate the appropriate transcriptional programme in cultured cells. Here, we developed a novel CRISPR activation system to induce the expression of genes that are expressed at low levels in cells that are generated in vitro, to explore the downstream consequences of their activation and to assess the effects on emerging haematopoietic progenitor cells.

When we compared endothelial cells derived in vitro from hiPSCs to those in the AGM region of the human embryo, at the time point when early haematopoietic commitment takes place, we identified nine transcription factors that were expressed at a lower level in the cultured cells. Some of these had been associated with blood cell development, including *GATA2* (*Castaño et al., 2019*; *de Pater et al., 2013*; *Ling et al., 2004*), SMAD7 (*McGarvey et al., 2017*), NR4A1 (*McGarvey et al., 2017*), SOX6 (*McGrath et al., 2011*), and RUNX1T1 (*Zeng et al., 2019*; *Calvanese et al., 2022*). Other genes such as ZNF124, ZNF33A, NFAT5, and TFDP2 had not been previously associated with haematopoiesis and could provide early evidence of their possible role in haematopoiesis that will require further studies. We were particularly interested in *RUNX1T1* (also known as ETO) which is associated with the leukemic fusion protein, AML1/ETO, resulting from the t(8;21) chromosomal translocation (*Rejeski et al., 2021*). Furthermore, *RUNX1T1* expression has been recently detected in transcriptomic analyses of the human AGM region (*Zeng et al., 2019*; *Calvanese et al., 2022*), but its precise

role during the ontogeny of the blood system has not been elucidated. The addition of a capture sequence to the gRNA backbone enabled their detection coincidentally with the single-cell transcriptome, and this allowed us to demonstrate that RUNX1T1 gRNAs were significantly enriched in cells within an expanded arterial cluster and their presence was associated with the highest expression of *IGFBP2* across clusters. *IGFBP2* KO mice show increased expression of cell cycle inhibitors and HSC apoptosis, implicating IGFBP2 as a modulator of HSCs cell cycle and survival (*Huynh et al., 2011*). More recently *IGFBP2* was reported as being highly expressed in the human AGM region at CS14 when HSCs are emerging (*Calvanese et al., 2022*), supporting a possible role during developmental haematopoiesis but the precise molecular process was unclear. In this study, we show that the addition of IGFBP2 recombinant protein in our in vitro model of EHT results in the emergence of an increased number of functional haematopoietic progenitors, and we provide an early observation to suggest that RUNX1T1 could be involved in the regulation of its expression. Because RUNX1T1 lacks a DNA binding domain, its direct involvement in the regulation of IGFBP2 expression, or in the development of the haematopoietic system in general, must require association with other cofactors that are yet to be identified.

Our data, together with in vivo data (*Calvanese et al., 2022*), show that IGFBP2 is expressed predominantly by endothelial cells. The supportive role of the endothelial niche in the development of HSCs has been studied in vivo (*Crosse et al., 2020*; *Hadland et al., 2022*; *Hadland et al., 2015*) and exploited in vitro to support HSCs emergence (*Hadland et al., 2022*; *Sandler et al., 2014*).

We describe here a novel role for IGFBP2 in the remodelling of the metabolism of iPSC-derived endothelial cells and that this is associated with the induction of RUNX1, a hallmark of haemogenic fate. The addition of IGFBP2 in the culture induces upregulation of the genes of the oxidative phosphorylation pathway in association with an increased relative mitochondrial contribution to cellular ATP production. This is due to reduced glycolysis in association with the downregulation of genes coding for glycolytic enzymes and their checkpoints. The switch between glycolytic to mitochondrial metabolism has been shown to be essential for the EHT in definitive haematopoiesis (*Oburoglu et al., 2022*; *Azzoni et al., 2021*). This switch is induced in vivo by mechanical cues downstream of the establishment of circulation, and it is required for functional HSCs development (*Azzoni et al., 2021*), while in vitro, the switch is driven by pyruvate mitochondrial catabolism, leading to definitive EHT as opposed to primitive (*Oburoglu et al., 2022*). Further in vivo studies are required to characterise the role of IGFBP2 during AGM haematopoiesis to overcome the limitation of using an in vitro model such as iPSCs differentiation.

In conclusion, we detected transcriptional differences between in vivo and in vitro developmental haematopoiesis and developed a novel inducible gene activation system to identify novel molecular players during the EHT transition. Our multidisciplinary approach identified IGFBP2 as novel signalling molecules that support human blood progenitor development in vitro, inducing a metabolic switch from cytoplasmic glycolysis to mitochondrial respiration. This study demonstrates that combining CRISPR-mediated activation of target genes with single-cell transcriptomic analysis in differentiating hPSCs can be a powerful approach to alter cell fate, providing a tool for gene function studies during human development. The fine epigenetic manipulation of the transcription can be readily applied to any cell lineage simply by adding specific gRNAs and it will be instrumental in exploring other developmental processes that can be, at least partially, mimicked in vitro with hiPSCs. Some limitations remain in applying CRISPR-mediated gene activation strategies in long differentiation protocols due to the challenge of detecting high copy numbers of the gRNAs, limiting the possibility of providing the fine statistical correlation needed to predict downstream target genes. Testing different promoters driving the gRNA expression or structural modifications of the gRNA scaffold could result in a more robust expression and allow for correlation analysis.

Finally, using this approach, we have identified the supportive role of IGFBP2, predominantly produced by endothelial cells, which induces transcriptional and metabolic remodelling in association with the induction of *RUNX1* expression, and result in higher haematopoietic progenitors' activity.

# Methods

## Resource availability

R code is available at https://github.com/afidanza/CRISPRa (copy archived at *afidanza, 2022*), and in the Source Code 1 file. Raw and processed data have been deposited to ArrayExpress (E-MTAB-12748). The AAVS1-iSAM are available on Addgene (RRID:Addgene_211495 ) and the gRNA 2.1 plasmids (RRID:Addgene_211496). Further information and requests for resources and reagents should be directed to and will be fulfilled by the corresponding author.

## Pluripotent stem cells maintenance

hPSCs were maintained in vitro in StemPro hESC SFM (Gibco) with bFGF (R&D) at 20 ng/ml. Wells were coated with Vitronectin (Thermo Fisher Scientific) at least 1 hr before plating and cells were passaged using the StemPro EZPassage tool (Thermo Fisher Scientific). Media change was performed every day and cells passaged every 3–4 days at a ratio of 1:4.

## Transfection

iPSCs SFCi55 and hESCs RUNX1-GFP were plated at $3 \times 10^5$ cells per well of a six-well plate and reverse transfected with 2 μg of DNA using the Xfect Transfection reagent (Clontech) and analysed 2 days later.

HeLa cells were cultured in Dulbecco's Modified Eagle Medium/Nutrient Mixture F-12 (DMEM/F12) with Glutmax and 5% FCS (Gibco) and passaged every few days at a ratio of 1:6. HEL were cultured in Iscove's Modified Dulbecco's Medium (IMDM) with 10% FCS (Gibco) and passaged every few days, at a ratio of 1:4. $2 \times 10^5$ cells were plated, transfected at 6–8 hr with 0.75 μg of DNA using Xfect Transfection reagent (Clontech), and then analysed 2 days after.

## Immunocytochemistry

Cells were fixed in 4% PFA in PBS at room temperature for 10 min, permeabilised in PBS-T (0.4% Triton X-100) for 20 min and blocked in PBS-T with 1% BSA and 3% goat serum for 1 hr. Primary antibodies were incubated in blocking solution overnight at 4°C (RUNX1 1:200 - ab92336, Abcam). Cells were then washed in PBS-T and incubated with secondary antibodies for 1 hr at room temperature (donkey α-rabbit 1:200 - A-11008 - Thermo Scientific). Cells were washed in PBS-T and counterstained with DAPI. Images were generated using the Zeiss Observer microscope.

## Gene expression analysis

Total RNA was purified using the RNAeasy Mini Kit (QIAGEN) and cDNA synthesised from 500 ng of total RNA using the High Capacity cDNA synthesis Kit (Applied Biosystems). 2 ng of cDNA was amplified per reaction and each reaction was performed in triplicate using the LightCycler 384 (Roche) with SYBR Green Master Mix II (Roche). A melting curve was performed and analysed for each gene to ensure the specificity of the amplification. *β-Actin* was used as reference genes to normalise the data (*Fidanza et al., 2017*).

## Pluripotent stem cells differentiation to haematopoietic progenitors

hPSCs were differentiated in a xeno-free composition of SFD medium (*Fidanza et al., 2020*), BSA was substituted with human serum albumin, HSA (Irvine-Scientific). Day 0 differentiation medium, containing 10 ng/ml BMP4, was added to the colonies prior to cutting. Cut colonies were transferred to a Cell Repellent 6 wells Plates (Greniner) to form embryoid bodies (EBs) and cultured for 2 days. At day 2 media was changed and supplemented with 3 μM CHIR (StemMacs). At day 3, EBs were transferred into fresh media supplemented with 5 ng/ml bFGF and 15 ng/ml VEGF. At day 6 media was changed for final haematopoietic induction in SFD medium supplemented with 5 ng/ml bFGF, 15 ng/ml VEGF, 30 ng/ml IL3, 10 ng/ml IL6, 5 ng/ml IL11, 50 ng/ml SCF, 2 U/ml EPO, 30 ng/ml TPO, 10 ng/ml FLT3L, and 25 ng/ml IGF1. From day 6 onward, cytokines were replaced every 2 days.

## CD34 isolation

CD34+ cells were isolated using CD34 Magnetic Microbeads from Miltenyi Biotec, according to their manufacturing protocol. Briefly, EBs were dissociated using Accutase (Life Technologies) at 37°C for

30 min. Cells were centrifuged and resuspended in 150 µl of PBS+0.5% BSA+2 mM EDTA with 50 µl Fcr blocker and 50 µl of magnetic anti-CD34 at 4°C for 30 min. Cells were washed using the same buffer and transferred to pre-equilibrated columns, washed three times, and eluted. After centrifugation, cells were resuspended in SFD media, counted and plated for OP9 coculture.

## OP9 coculture and colony assay

OP9 cells were maintained in α-MEM supplemented with 20% serum (Gibco) and sodium bicarbonate (Gibco) and passaged with trypsin every 3–4 days. The day before the coculture, 45.000 OP9 cells were plated for each 12-well plates' well in SFD media. The day of the coculture the 20.000 iSAM cells or 25.000 SFCi55 or H9 were plated in each well and cultured in SFD media supplemented with 5 ng/ml bFGF, 15 ng/ml VEGF, 30 ng/ml IL3, 10 ng/ml IL6, 5 ng/ml IL11, 50 ng/ml SCF, 2 U/ml EPO, 30 ng/ml TPO, 10 ng/ml FLT3L, and 25 ng/ml IGF1 and 100 ng/ml IGFBP2. Cytokines were replaced twice during 1 week of coculture. At the end of the coculture, cells were collected by trypsin and half of the well equivalent was plated in 2 ml of methylcellulose medium (human enriched H4435, STEMCELL Technologies). Cells were incubated in the assay for 14 days and then scored.

## Laminin EHT culture

Laminin EHT culture was performed as previously described (*Ventura et al., 2023*). Briefly, 24-well plates were coated for at least 2 hr with recombinant human Laminin-521 (Thermo Fisher). Following CD34+ isolation at day 8, 400.000 CD34+ cells were seeded in each well of a precoated 24-well plate in SFD media supplemented with 5 ng/ml bFGF, 15 ng/ml VEGF, 30 ng/ml IL3, 10 ng/ml IL6, 5 ng/ml IL11, 50 ng/ml SCF, 2 U/ml EPO, 30 ng/ml TPO, 10 ng/ml FLT3L, and 25 ng/ml IGF1 and 100 ng/ml IGFBP2.

## Flow cytometry staining and cell sorting

EBs were dissociated using Accutase (Life Technologies) at 37°C for 30 min. Cells were centrifuged and resuspended in PBS+0.5% BSA+2 mM EDTA, counted and stained at $10^5$ cells for a single tube. Cells were stained with antibodies for 30 min at room temperature with gentle shaking. Flow cytometry data were collected using DIVA software (BD).

For the sorting experiments, the cells were prepared as above and stained at $10^7$ cells/ml in the presence of the specific antibodies. Sorting was performed using FACSAria Fusion (BD) and cells were collected in PBS+1% BSA. Data were analysed using FlowJo version 10.4.2.

## Flow cytometry antibodies

For flow cytometry $10^5$ cells per test were stained in 50 µl of staining solution with the following antibodies: CD34 Percp-Efluor710 (4H11 eBioscience, 1:100), CD34 Pe (4H11 eBioscience, 1:200), CD43 APC (eBio84-3C1, 1:100), CD45 FITC (2D1 eBioscience, 1:100), DLL4 Pe (MHD4-46 BioLegend, 1:200), CD41 PE (HIP8 BioLegend, 1:200), CD144 APC (16B1 eBioscience, 1:100), CD235a FITC (HIR2 BD Bioscience, 1:250).

## iSAM plasmid generation

The iSAM plasmid was obtained by Gibson assembly of four fragments. The first fragment, the backbone, was a DOX-inducible AAVS1-targeted plasmid expressing an E6-E7-IRES-ZsGreen which was excised by BstBI and NdeI. The second fragment, one of the adapters, was derived from the UniSAM plasmid that we previously generated (RRID:Addgene_99866) by PCR with the following primer sets FW_aggggacccggttcgagaaggggctcttcatcactagggccgctagctctagagagcgtcgaatt, RV_ttcgggtcccaattgc cgtcgtgctggcggctcttcccacctttctcttcttcttggggctcatggtggcc. The UniSAM cassette was also obtained from the UniSAM plasmid via digestion with BsrGI and BsiWI. Finally, the last fragment consisting of another adapter for the Gibson was custom synthetised and contained overlapping sequences flanking a chicken β-globin insulator that we inserted to prevent cross-activation of the EF1α promoter and the TRE-GV promoter driving the iSAM. Correct assembly was verified by Sanger sequencing.

## iSAM cell lines derivation

The iSAM plasmid was used together with ZNFs specific for the AAVS1 locus to mediate specific integration in SFCi55 hiPSCs line (*Fidanza et al., 2020*; *Yang et al., 2017*). Briefly, 10 µg of AAVS1-iSAM

plasmid with 2.5 µg of each ZNFs plasmid, left and right, were co-transfected using Xfect (Takara) according to the manufacturer's protocol. Cells were selected using Neomycin. Single clones were picked, amplified, and initially screened by mCherry expression upon DOX addition. Clones that expressed the fluorescent tag were screened for specific integration using PCR followed by Sanger sequencing for the correctly integrated clones. 100 ng of genomic DNA was amplified using the EmeraldAmp MAX HS Takara and specific primer sets (*Supplementary file 1*). For the specific AAVS1 integration site, Sigma_AAVS1 - CGG AAC TCT GCC CTC TAA CG and NeoR - GAT ATT GCT GAA GAG CTT GGC GG were used with the PCR conditions of 95°C for 7 min, 32 cycles of 95°C for 15 s, 57°C for 30 s, and 72°C for 1 min, with the final elongation step at 72°C for 7 min. For the wild-type locus screening, Sigma_AAVS1 - CGG AAC TCT GCC CTC TAA CG and AAVS1_EXT3_RV - ACA CCC AGA CCT GAC CCA AA were used with the PCR cycling conditions of 95°C for 7 min, 30 cycles of 95°C for 15 s, 57°C for 30 s, and 72°C for 2 min, with the final elongation step at 72°C for 6 min.

## Capture sequencing addition to the gRNA backbone

The capture sequence 2 was added to the gRNA_Puro_Backbone (RRID:Addgene_73797) by PCR. Briefly, the capture sequence was added before the termination signal of the gRNA followed by a BamHI site using the following PCR primers: gRNA_FW gagggcctatttcccatgattcct, gRNA_Cap_RV aaaaaaggatccaaaaaaCCTTAGCCGCTAATAGGTGAGCgcaccgactcggtgcc.

The gRNA backbone was replaced from the original plasmid via NdeI and BamHI digestion, followed by ligation of the PCR produced following the same digestion. Correct integration of the insert was verified by Sanger sequencing.

## Target genes identification

Candidate genes were identified by comparing in vivo haemogenic endothelium (*Zeng et al., 2019*) (GSE135202) and in vitro iPSCs-derived endothelial cells that we previously generated (*Fidanza et al., 2020*) (E-MTAB-9295). Briefly, the two datasets were merged and normalised using the R package Seurat. Specific markers for haemogenic endothelium were identified and transcription factors were sorted based on their GO annotation. Within those genes we filtered those detected in more than 50% of the in vivo haemogenic endothelium and expressed in less than 25% of the in vitro-derived endothelial cells. This pipeline identified nine target genes *RUNX1T1*, *NR4A1*, *GATA2*, *SMAD7*, *ZNF124*, *SOX6*, *ZNF33A*, *NFAT5*, *TFDP*.

## AGM gRNAs library preparation

sgRNA design was performed by selecting the top candidates for on-target and off-target score. Between five and seven guides per gene were designed for *RUNX1T1*, *NR4A1*, *GATA2*, *SMAD7*, *ZNF124*, *SOX6*, *ZNF33A*, *NFAT5*, *TFDP2* using the CRISPRpick tool from the Broad Institute (https://portals.broadinstitute.org/gppx/crispick/public) (*Supplementary file 1*). All the guide variants were Golden Gate cloned with the gRNA 2.1 backbone according to the established protocol (*Konermann et al., 2015*). The 49 plasmids were pooled together in an equimolar ratio and the library prep was subsequently used to produce lentiviral particles with a second-generation production system. Briefly, the psPAX2 packaging plasmid, pMD2.G envelope, and the AGM vector library were co-transfected using polyethyleneimine (Polysciences, Warrington, PA, USA) as previously detailed (*Petazzi et al., 2020*). Lentiviral particles-containing supernatants were harvested 48–72 hr post-transfection, concentrated by ultracentrifugation, and titered in hiPSCs cells.

## iSAM_AGM and iSAM_NT cell line derivation

The selected iSAM clone (3.13 internal coding) was infected with viral particles containing either the AGM library or the non-targeting gRNA (NT) at a MOI of 10. The iSAM cell line was plated the afternoon before at 7×10⁶ cells into a T125 in the presence of 10 µM Rock Inhibitor (Merk) which was maintained until the day following the infection. Cells were infected in the presence of 8 µg/ml of Polybrene (Merk). Puromycin selection was initiated 36 hr post-infection and maintained during their culture until the beginning of the differentiation. Both lines were tested for integration of the gRNAs. Briefly, 100 ng of isolated gDNA was amplified using the PrimeSTAR MAX PCR mix (Takara) using the primers gRNA_screening_FW and gRNA_screening_RV (*Supplementary file 1*). Purified amplicons were subjected to Sanger sequencing.

## Single-cell RNA sequencing

For the iSAM_AGM and iSAM_NT scRNAseq experiment, EBs obtained from day 10 of differentiation were dissociated using Accutase (Life Technologies) at 37°C for 30. For the IGFBP2 experiment, day 10 and day 13 cells from the Laminin EHT culture were detached from the adherent layer using Accutase (Life Technologies) at 37°C for 5'. From both experiments, cells were centrifuged and resuspended in CD34-Pe staining solution at a density of $10^7$/ml. CD34+/live/single cells were FAC-sorted in PBS+0.1% BSA. Cell viability was also confirmed by Trypan blue stain for an accurate count. Around 15,000 cells per sample were loaded into the 10X Chromium Controller, and single-cell libraries were obtained using the Chromium Single Cell 3' Reagent Kits v3 (10X Genomics) according to the manufacturer's protocol. The four libraries were indexed using SI PCR primers with different i7 indexes to allow for demultiplexing of the sequencing data. RNA concentration was obtained using Quibit RNA HS (Thermo Fisher). Quality of the obtained libraries were verified using LabChip GX (Perkin-Elmer). Libraries were sequenced using NextSeq 2000 technology (Illumina) at 50.000 reads/cell. Data were aligned to GRCh38 using the Cell Ranger dedicated pipeline (10X Genomics). Data filtering, dimension reduction, clustering analysis, differentially expressed genes, and cell cycle analysis were obtained using Seurat R package (version 4.1.0) (*Hao et al., 2021*). Cells were subjected to QC and filtering using both the number of genes (1000–7500) and the percentage of mitochondrial genes detected (1–15%), resulting in 9025, 10,942, 9468, 13,073 cells respectively for the samples iSAM_NT, iSAM_NT+DOX, iSAM_AGM, and iSAM_AGM+DOX. Pseudotemporal ordering was performed using Monocle 3 R package (*Cao et al., 2019*). KEGG pathways was performed using ShinyGo (*Ge et al., 2020*). The gRNAs' expression matrix was used to select cells in which the expression of the gRNAs was detected. Briefly, for the libraries derived from the iSAM_NT control containing only the NT gRNA, the filter was set for cells expressing one gRNA, while for the iSAM_AGM libraries was set to more than one. The code is available on GitHub at https://github.com/afidanza/CRISPRa, copy archived at *afidanza, 2022* the raw data have been submitted to Array Express (E-MTAB-12748), and the browsable processed data will be made available at the time of publication on our website containing previous sequencing data at https://lab.antonellafidanza.com.

## ATP production analysis

At day 8 of differentiation, 20.000 CD34+ were plated for each precoated well of the Seahorse XFp Mini Cell Culture Plates (Agilent) precoated with rhLaminin-521 (Thermo Fisher). On day 13, the cells' ATP production rate was analysed with the Seahorse XF Real-Time ATP Rate Assay kit (Agilent) according to the manufacturer's protocol, following confirmation of comparable cell densities across replicates and conditions. Briefly, the media was changed prior to the assay for the XF DMEM, pH 7.4, supplemented with 10 mM Seahorse XF glucose, 1 mM Seahorse XF pyruvate and 2 mM Seahorse XF Glutamine (Agilent) and incubated for 45–60 min in a non-$CO_2$ incubator. Oligomycin and Rotenone/AA solutions were prepared and added in the cartridge and finally loaded together with the cells in the Seahorse XF Mini Analyzer using the dedicated software. Data were collected at the end of the run, and the values of ATP production were calculated according to the ATP Production Rate Calculation provided by Agilent. Briefly, OCR ATP was calculated as $OCR_{basal}$-$OCR_{oligo}$ averaged across the three reads for each well. Mitochondrial ATP was calculated as $OCR_{ATP}$ multiplied by the molecular oxygen consumption rate of 2 and by the P/O value of 2.75. For the glycolytic ATP production, we calculated the MitoPER as the $OCR_{basal}$-$OCR_{rot}$ times the $CO_2$ contribution factor of 0.5. The PER was then calculated as the ECAR times the buffer factor of 2.6, the volume of reaction of 2.28 µl, and the Kvol value of 1.1. The GlycoATP production rate was obtained by removing the MitoPER from the PER, and the TotalATP was calculated by adding the GlycoATP and the MytoATP. Mito/Glyco ratio was obtained by dividing their ATP production value.

## Cell cycle analysis

DAPI staining and flow cytometry analysis were performed to verify the proliferation rate of the cells. Briefly, cells were collected from the supernatant by aspiration using a pastette, washed in PBS+0.5% BSA+2 mM EDTA, counted and stained at $10^5$ cells for a single tube. Cells were stained 1:1 vol:vol with a solution of 1% NP40 and 5 µg/ml DAPI for 2 min and acquired using the DIVA software (BD), and analysed using FlowJo version 10.4.2.

## Acknowledgements

AF and LF acknowledge financial support from the Biotechnology and Biological Sciences Research Council; Grant S002219/1. AF was supported by a European Hematology Association Advanced Research Grant (EHA RAG 2021), and by the American Society for Hematology (Research Global Award). TV and AM were supported by PhD studentships from the Medical Research Council (Precision Medicine) and the College of Medicine and Veterinary Medicine, respectively. FPL was supported by an Erasmus+ Traineeship Program 2016/2017. PM acknowledges financial support from a PERIS program from the Catalan Government and a Retos collaboration project from the MINECO (RTC-2018-4603-1)

## Additional information

### Funding

| Funder | Grant reference number | Author |
|---|---|---|
| Biotechnology and Biological Sciences Research Council | S002219/1 | Lesley M Forrester Antonella Fidanza |
| European Hematology Association | EHA RAG 2021 | Antonella Fidanza |
| American Society for Hematology | Research Global Award | Antonella Fidanza |
| Medical Research Council | Precision Medicine PhD scholarship | Telma Ventura |
| College of Medicine and Veterinary Medicine, University of Edinburgh | Tissue Repair PhD studentship | Alisha May |
| Erasmus + | Traineeship Program 2016/2017 | Francesca Paola Luongo |
| Catalan Government | PERIS program | Paolo Petazzi Pablo Menendez |
| MINECO | RTC-2018-4603-1 | Paolo Petazzi Pablo Menendez |

The funders had no role in study design, data collection and interpretation, or the decision to submit the work for publication.

### Author contributions

Paolo Petazzi, Alisha May, Investigation, Writing – review and editing; Telma Ventura, Francesca Paola Luongo, Heather McClafferty, Helen Alice Taylor, Investigation; Michael J Shipston, Investigation, Writing – review and editing, Resources; Nicola Romanò, Resources, Data curation, Formal analysis, Investigation, Methodology, Writing – review and editing; Lesley M Forrester, Supervision, Funding acquisition, Investigation, Writing – original draft, Project administration, Writing – review and editing; Pablo Menendez, Methodology, Writing – review and editing; Antonella Fidanza, Conceptualization, Resources, Data curation, Formal analysis, Supervision, Funding acquisition, Investigation, Methodology, Writing – original draft, Project administration, Writing – review and editing

### Author ORCIDs

Telma Ventura http://orcid.org/0009-0007-2356-2208
Francesca Paola Luongo https://orcid.org/0000-0001-6286-5331
Alisha May https://orcid.org/0000-0003-2101-6047
Michael J Shipston https://orcid.org/0000-0001-7544-582X
Nicola Romanò https://orcid.org/0000-0002-1316-8940
Pablo Menendez https://orcid.org/0000-0001-9372-1007
Antonella Fidanza https://orcid.org/0000-0002-1266-6454

Reviewer #1 (Public Review): https://doi.org/10.7554/eLife.94884.3.sa1
Author response https://doi.org/10.7554/eLife.94884.3.sa2

## Additional files

### Supplementary files

• Supplementary file 1. Spreadsheet containing tables of the gRNAs' sequence used to activate the target genes and the non-targeting gRNA used as control (tab 1), the marker genes identified for the cell clusters identified in *Figure 3* (tab 2), the differentially expressed genes identified from the comparison of in vivo aorta-gonad-mesonephros (AGM) cells with those derived in vitro from human induced pluripotent stem cells (iPSCs) (tab 3), the gRNA enrichment analysis in arterial cells iSAM_AGM+DOX versus iSAM_AGM-DOX (tab 4).

• MDAR checklist

### Data availability

The source code has also been deposited on GitHub https://github.com/afidanza/CRISPRa (copy archived at *afidanza, 2022*). The plasmids are available on AddGene RRID:Addgene_211495 and RRID:Addgene_211496. The raw and processed RNA seq data have been deposited on ArrayExpressed Accession number E-MTAB-12748. The analysed data can be browsed on our online web tool at https://lab.antonellafidanza.com/.

The following dataset was generated:

| Author(s) | Year | Dataset title | Dataset URL | Database and Identifier |
|---|---|---|---|---|
| Fidanza A, Romano N | 2024 | Single-cell RNAseq of differentiating human iPSCs during endothelial-to-hematopoietic transition with targeted CRISPR activation | https://www.ebi.ac.uk/biostudies/arrayexpress/studies/E-MTAB-12748?query=E-MTAB-12748 | ArrayExpress, E-MTAB-12748 |

The following previously published dataset was used:

| Author(s) | Year | Dataset title | Dataset URL | Database and Identifier |
|---|---|---|---|---|
| Zeng Y, He J, Bai Z, Li Z, Lan Y, Liu B | 2019 | Tracing the first hematopoietic stem cell generation in human embryo by single-cell RNA sequencing | https://www.ncbi.nlm.nih.gov/geo/query/acc.cgi?acc=GSE135202 | NCBI Gene Expression Omnibus, GSE135202 |

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

# Appendix 1

## Appendix 1—key resources table

| Reagent type (species) or resource | Designation | Source or reference | Identifiers | Additional information |
|---|---|---|---|---|
| Strain, strain background (*Escherichia coli*) | StBL3 | Expanded in the lab | StBL3 | Electrocompetent cells for the expansion and cloning of the plasmid in this paper, use of alternative strains leads to lower yield |
| Cell line (*Homo sapiens*) | HeLa | Lab | HeLa | Epithelial immortalised cell line from cervical cancer |
| Cell line (*Homo sapiens*) | hiPSCs | Lab, reference: Yang CT, et al. *Stem Cells*. 2017;35(4):886–897. | SFCi55 | Dermal fibroblast from O Rh- donor reprogrammed using episomal *OCT4*, *KLF2*, *SOX2*, and *MYC* |
| Cell line (*Homo sapiens*) | hiPSCs | This paper | iSAM | Human iPSCs with the DOX-inducible SAM cassette inserted in the AAVS1 locus |
| Cell line (*Homo sapiens*) | hiPSCs | This paper | iSAM-AGM | Human iPSCs with the DOX-inducible SAM cassette inserted in the AAVS1 locus+random integration of the 49 gRNAs |
| Cell line (*Homo sapiens*) | hiPSCs | This paper | iSAM-NT | Human iPSCs with the DOX-inducible SAM cassette inserted in the AAVS1 locus+random integration of the non-targeting gRNA |
| Recombinant DNA reagent | AAVS1-iSAM (plasmid) | This paper | RRID:Addgene_211495 | AAVS1 targeting vector for the insertion of the iSAM cassette on chromosome 19 |
| Recombinant DNA reagent | gRNA 2.1 backbone (plasmid) | This paper | RRID:Addgene_211496 | Lentiviral vector for the U6-driven expression of gRNA compatible with the SAM system containing the capture sequencing for 10X scRNAseq |
| Recombinant DNA reagent | Non-targeting CTR gRNA | This paper, *Supplementary file 1* | NT-gRNA | CGGAGGCTAAGCGTCGCAAC |
| Recombinant DNA reagent | Library of targeting gRNA | This paper, *Supplementary file 1* | AGM-gRNAs | |
| Antibody | Anti-human CD34 Percp-Efluor710 (mouse monoclonal) | eBioscience | Clone 4H11 | Flow cytometry 1:100 |
| Antibody | Anti-human CD34 PE (mouse monoclonal) | eBioscience | Clone 4H11 | Flow cytometry 1:200 |
| Antibody | Anti-human CD43 APC (mouse monoclonal) | eBioscience | Clone eBio84-3C1 | Flow cytometry 1:100 |
| Antibody | Anti-human CD45 FITC (mouse monoclonal) | eBioscience | Clone 2D1 | Flow cytometry 1:100 |
| Antibody | Anti-human DLL4 PE (mouse monoclonal) | BioLegend | Clone MHD4-46 | Flow cytometry 1:200 |
| Antibody | Anti-human CD41 PE (mouse monoclonal) | BioLegend | Clone HIP8 | Flow cytometry 1:200 |
| Antibody | Anti-human CD144 APC (mouse monoclonal) | eBioscience | Clone 16B1 | Flow cytometry 1:100 |
| Antibody | Anti-human CD235a FITC (mouse monoclonal) | BD Bioscience | Clone HIR2 | Flow cytometry 1:250 |
| Sequence-based reagent | FW | Synthetised by IDT | PCR primer for the amplification of the UniSAM plasmid RRID:Addgene_99866 | aggggacccggttcgaga aggggctcttcatcactagg gccgctagctctagagagcgtcgaatt |
| Sequence-based reagent | RV | Synthetised by IDT | PCR primer for the amplification of the UniSAM plasmid RRID:Addgene_99866 | ttcgggtcccaattgccgtcg tgctggcggctcttcccaccttttctc ttcttcttggggctcatggtggcc |

*Appendix 1 Continued on next page*

*Appendix 1 Continued*

| Reagent type (species) or resource | Designation | Source or reference | Identifiers | Additional information |
|---|---|---|---|---|
| Sequence-based reagent | Sigma_AAVS1 | Synthetised by IDT | Forward PCR primer to test for the specific AAVS1 integration of the iSAM cassette and for the wild type not integrated locus | CGG AAC TCT GCC CTC TAA CG |
| Sequence-based reagent | NeoR | Synthetised by IDT | Reverse PCR primer to test for the specific AAVS1 integration of the iSAM cassette | GAT ATT GCT GAA GAG CTT GGC GG |
| Sequence-based reagent | AVVS1_EXT3_RV | Synthetised by IDT | Forward PCR primer to test for the wild type AVVS1 not integrated locus | ACA CCC AGA CCT GAC CCA AA |
| Sequence-based reagent | gRNA_FW | Synthetised by IDT | Forward primer used for the PCR to insert the capture sequence in the gRNA plasmid RRID:Addgene_73797 | gagggcctatttcccatgattcct |
| Sequence-based reagent | gRNA_Cap_RV | Synthetised by IDT | Reverse primer used for the PCR to insert the capture sequence in the gRNA plasmid RRID:Addgene_73797 | aaaaaaggatccaaaaaa CCTTAGCCGCTAATAG GTGAGCgcaccgactcggtgcc. |
| Peptide, recombinant protein | rhBMP4 | R&D | 314-BP-010 | 20 ng/ml |
| Peptide, recombinant protein | rhEPO | R&D | 287-TC-500 | 3 U/ml |
| Peptide, recombinant protein | rhIGF1 | Peprotech | 100-11-100uG | 25 ng/ml |
| Peptide, recombinant protein | rhIL11 | Peprotech | 200-11-10uG | 5 ng/ml |
| Peptide, recombinant protein | rhIL3 | Peprotech | 200-03-10uG | 30 ng/ml |
| Peptide, recombinant protein | rhIL6 | R&D | 206-IL-010 | 10 ng/ml |
| Peptide, recombinant protein | rhSCF | Life Technologies | PHC2111 | 50 ng/ml |
| Peptide, recombinant protein | rhTPO | R&D | 288-TPN-025 | 30 ng/ml |
| Peptide, recombinant protein | rhVEGF | R&D | 293-VE-010 | 15 ng/ml |
| Peptide, recombinant protein | rhIGFBP2 | BioLegend | 793404 | 100 ng/ml |
| Chemical compound, drug | CHIR | Cayman | 13122-1mg-CAY | 3 µM |

