## [Editor Report · eLife Assessment]

This study presents **useful** findings to inform and improve the in vitro differentiation of hematopoietic progenitor cells from human induced pluripotent stem cells. Relying on a well-characterised technical approach, the data analysis is overall **solid** and reasonably supports the main conclusions.

---

## [Referee Report · Reviewer #1 (Public Review)]

Summary:

The work from Petazzi et al. aimed at identifying novel factors supporting the differentiation of human hematopoietic progenitors from induced-pluripotent stem cells (iPSCs). The authors developed an inducible CRISPR-mediated activation strategy (iCRISPRa) to test the impact of newly identified candidate factors on the generation of hematopoietic progenitors in vitro. They first compared previously published transcriptomic data of iPSC-derived hemato-endothelial populations with cells isolated ex vivo from the aorta-gonad-mesonephros (AGM) region of the human embryo and they identified 9 transcription factors expressed in the aortic hemogenic endothelium that were poorly expressed in the in vitro differentiated cells. They then tested the activation of these candidate factors in an iPSC-based culture system supporting the differentiation of hematopoietic progenitors in vitro. They found that the IGF binding protein 2 (IGFBP2) was the most upregulated gene in arterial endothelium after activation and they demonstrated that IGFBP2 promotes the generation of functional hematopoietic progenitors in vitro.

Strengths:

The authors developed a very useful doxycycline-inducible system to activate the expression of specific candidate genes in human iPSC. This approach allows us to simultaneously test the impact of 9 different transcription factors on in vitro differentiation of hematopoietic cells, and the system appears to be very versatile and applicable to a broad variety of studies. Using this approach, the authors exhaustively demonstrated the role of IGFBP2 in promoting the generation of functional hematopoietic progenitors in vitro from iPSCs.

Weaknesses:

The authors performed a very thorough characterization of the system in proof-of-principle experiments activating a single transcription factor. However, when 9 independent factors were used, it is not always clear whether the observed results were the consequence of the simultaneous activation of all 9 TF or just a subset of them.

---

## [Author Response]

The following is the authors’ response to the original reviews.

**Public Reviews:**

**Reviewer #1 (Public Review):**
Summary:The work from Petazzi et al. aimed at identifying novel factors supporting the differentiation of human hematopoietic progenitors from induced pluripotent stem cells (iPSCs). The authors developed an inducible CRISPR-mediated activation strategy (iCRISPRa) to test the impact of newly identified candidate factors on the generation of hematopoietic progenitors in vitro. They first compared previously published transcriptomic data of iPSCderived hemato-endothelial populations with cells isolated ex vivo from the aorta-gonadmesonephros (AGM) region of the human embryo and they identified 9 transcription factors expressed in the aortic hemogenic endothelium that were poorly expressed in the in vitro differentiated cells. They then tested the activation of these candidate factors in an iPSCbased culture system supporting the differentiation of hematopoietic progenitors in vitro. They found that the IGF binding protein 2 (IGFBP2) was the most upregulated gene in arterial endothelium after activation and they demonstrated that IGFBP2 promotes the generation of functional hematopoietic progenitors in vitro.Strengths:The authors developed an extremely useful doxycycline-inducible system to activate the expression of specific candidate genes in human iPSC. This approach allows us to simultaneously test the impact of 9 different transcription factors on in vitro differentiation of hematopoietic cells, and the system appears to be very versatile and applicable to a broad variety of studies.The system was extensively validated for the expression of 1 transcription factor (RUNX1) in both HeLa cells and human iPSC, and a detailed characterization of this test experiment was provided.The authors exhaustively demonstrated the role of IGFBP2 in promoting the generation of functional hematopoietic progenitors in vitro from iPSCs. Even though the use of IGFBP2interacting proteins IGF1 and IGF2 have been previously reported in human iPSC-derived hematopoietic differentiation in vitro (Ditadi and Sturgeon, Methods 2016; Ng et al., Nature Biotechnology 2016), and IGFBP-2 itself has been shown to promote adult HSC expansion ex vivo (Zhang et al., Blood 2008), its role on supporting in vitro hematopoiesis was demonstrated here for the first time.Weaknesses:Although the authors performed a very thorough characterization of the system in proof-ofprinciple experiments activating a single transcription factor, the data provided when 9 independent factors were used is not sufficient to fully validate the experimental strategy. Indeed, in the current version of the manuscript, it is not clear whether the results presented in both the scRNAseq analysis and the functional assays are the consequence of the simultaneous activation of all 9 TF or just a subset of them. This is essential to establish whether all the proposed factors play a role during embryonic hematopoiesis, and a more complete analysis of the scRNAseq dataset could help clarify this aspect.Similarly, the data presented in the manuscript are not sufficient to clarify at what stage of the endothelial-to-hematopoietic transition (EHT) the TF activation has an impact. Indeed, even though the overall increase of functional hematopoietic progenitors is fully demonstrated, the assays proposed in the manuscript do not clarify whether this is due to a specific effect at the endothelial level or to an increased proliferation rate of the generated hematopoietic progenitors. Similar conclusions can be applied to the functional validation of IGFBP2 in vitro.The overall conclusions are sometimes vague and not always supported by the data. For instance, the authors state that the CRISPR activation strategy resulted in transcriptional remodeling and a steer in cell identity, but they do not specify which cell types are involved and at what level of the EHT process this is happening. In the discussion, the authors also claim that they provided evidence to support that RUNX1T1 could regulate IGFBP2 expression. However, this is exclusively based on the enrichment of RUNX1T1 gRNA in cells expressing higher levels of IGFBP2 and it does not demonstrate any direct or indirect association of the two factors.

We thank the reviewer for the positive comments about the importance of our work and have now addressed the points raised as weaknesses by performing additional analysis and experiments, adding a new schematic of the mechanism, and rewording our claims.

We have clarified the different effects mediated by the activation and the IGFBP2 addition in a summary section at the end of the results and added Figure 6, showing this in visual form. We have also clearly stated the limitations related to the correlation between RUNX1T1 and IGFBP2 in the discussion and toned down our claims regarding this throughout the entire paper. We have also reworded the text to clarify the specific cell types identified in the sequencing data that we refer to.

**Reviewer #2 (Public Review):**
To enable robust production of hematopoietic progenitors in-vitro, Petazzi et al examined the role of transcription factors in the arterial hemogenic endothelium. They use IGFBP2 as a candidate gene to increase the directed differentiation of iPSCs into hematopoietic progenitors. They have established a novel induced-CRISPR mediated activation strategy to drive the expression of multiple endogenous transcription factors and show enhanced production of hematopoietic progenitors through expansion of the arterial endothelial cells. Further, upregulation of IGFBP2 in the arterial cells facilitates the metabolic switch from glycolysis to oxidative phosphorylation, inducing hematopoietic differentiation. While the overall study and resources generated are good, assertions in the manuscript are not entirely supported by the experimental data and some claims need further experimental validation.

We thank the reviewer for the positive comments, and we have provided new data and analysis to make sure that all our assertations are clearly supported and also reworded those where limitations were identified by the reviewers.

**Recommendations for the authors:**

**Reviewing Editor (Recommendations For The Authors):**
The assessment could change from "incomplete" to "solid" if the authors: (i) improve data analysis (for both scRNAseq and functional assays) by providing additional information that could strengthen their conclusions, as suggested in the specific comments by both reviewers; (ii) either provide new functional evidence supporting their mechanistic conclusion or alternatively tone down the claims that are not fully supported by data and acknowledge the limitations raised by reviewers in the discussion; (iii) the issue of paracrine signaling to expand only hematopoietic progenitors needs to be addressed.

We have now improved the data analysis and provided additional functional tests to strengthen our conclusions and toned down those that were identified by the reviewers as not supported enough and included a discussion on these limitations. We have also reworded the section about the paracrine signaling throughout the paper.

**Reviewer #1 (Recommendations For The Authors):**
Figure 1 contains exclusively published data. It might be more appropriate to use it as a supplementary figure or as part of a more exhaustive figure (maybe combining Figures 1 and 2 together?).

Figure 1 contained novel bioinformatic analyses that represent the base of our research and it has a different content and focus to figure 2, which is already a large figure. We therefore believe it is better to keep it as a separate figure, containing a new panel now too.

It seems there is an issue with Figure S3 labelling:• In line 112, Figure S2A-B does not display genomic PCR and sequencing results;• In line 123, Figure S3D-E does not show viability and proliferation data;• In line 127, Figure S3G does not show mCherry expression in response to DOX;

We apologies for the confusion with the numbers, we have now correctly labelled the figures.

It would be more informative to include gates and frequency on flow cytometry plots in Figure S3, to be able to evaluate the extent of the reduction in mCherry expression.

We have now included the gating and frequency of mCherry-expressing cells in Supplementary Figure 3D.

It is not clear from the text and figures whether the SB treatment was maintained throughout the hematopoietic differentiation protocol (line 122):• If so, it would be important to confirm that HDAC treatment does not affect EHT cultures• If not, can the authors provide some evidence that transgene silencing is not occurring during hematopoietic differentiation?

We have clarified that we decided to treat the cells with SB exclusively in maintenance condihons because HDACs have been shown to be essenhal for the EHT (lines 138-142). We have now also included addihonal data showing the high expression of the mCherry tag reporhng the iSAM expression on day 8 (Supplementary Figure 4F).

Can the authors provide a simple diagram summarizing the experimental strategy for each differentiation experiment in the respective supplementary figure? For instance, at what stage of the protocol was DOX added in Figure 3? Or at what stage IGFBP2 was added in Figure 5? It would be a very useful addition to the interpretation of the results.

We have now included three schemahcs for all the experiments in the manuscript in supplementary figure 4 A-C.

In Figure 3, the authors should provide more detailed information about the data filtering of the scRNAseq experiment, and more specifically:• How many cells were included in the analysis for each library after QC and filtering?• How "cells in which the gRNAs expression was detected" were selected? Do they include only cells showing expression of gRNAs for all 9 TF?

This informahon is now included in the method sechon lines 773-781; the detailed code is available on the GitHub link provided in the same sechon. We have filtered the cells expressing one gRNA for the non-targehng gRNA (iSAM_NT) control and more than one for the iSAM_AGM sample.

In Figure 3A, it is not clear whether the expression of the 9 factors is consistently detected in all cells or just a subset of them, and the heatmap in Figure 3A does not provide this information. It would be more accurate to provide expression on a per-cell basis, for instance, as a violin plot displaying single dots representing each cell.

We have now included this violin plot in Supplementary Figure 4G as requested. However, this visualisation is difficult to interpret because some of the target genes’ expression seems variable in both experimental and control conditions. We had envisaged that this could have been the case and so this is why we had included the three different controls. For this reason we chose to show the normalised expression which takes all the different variables into account (Figure 3A).

In Figure 3B-C, it seems that clusters EHT1 and EHT2 do not express endothelial markers anymore. Are these fully differentiated hematopoietic cells rather than cells undergoing EHT? In general, it would be quite important to provide evidence of expressed marker genes characterizing each cluster (eg. heatmap summarizing top DEG in the supplementary figure?).

We have now provided a spreadsheet containing the clusters’ markers that we used in

Supplementary Table 1 a heatmap in Figure 3E. Furthermor,e we have now edited Figure 3C to include Pan Endothelial markers (PECAM1 and CDH5). These data show that the EHT1 and EHT2 cluster both express endothelial markers but are progressively downregulated as expected during endothelial to hematopoietic transition. We have also included and discussed this in the manuscript lines 192-195 and a schematic for the mechanism in Figure 6.

In Figure 3E, displaying the proportion of clusters within each sample/library would be a more accurate way of comparing the cell types present in each library (removing potential bias introduced by loading different numbers of cells in each sample).

We have now included the requested data in Supplementary Figure 4I and it confirms again the expansion of arterial cells in the activated cells.

In Figure 3G, by plating 20,000 total CD34+, the assay does not account for potential differences in sample composition. It is then hard to discriminate between the increased number of progenitors in the input or an enhanced ability of HE to undergo EHT. This is an important aspect to consider to precisely identify at what level the activation of the 9 factors is acting. A proper quantification of flow cytometry data summarizing the % of progenitors, arterial cells, etc. would be useful to interpret these results.

Lines 204-205 reworded. We are very much aware of the fact that the CD34+ cell population consists of a range of cells across the EHT process and this is precisely why we carried out this single cell sequencing analyses. We purposely tested the effect of the observed changes in composition by colony assays

In Figure 3G, it seems that NT cells w/o DOX have very little CFU potential (if any). Can the authors provide an explanation for this?

We think that the limited CFU potential is due to the extensive genetic manipulation and selection that the cells underwent for the derivation of all the iSAM lines but this did not impede us from observing an effect of gene activation on CFU numbers. This is one of the primary reasons that we then validated our overall findings using the parental iPSC line in control condition and with the addition of IGFBP2. We show that the parental iPSC line gives rise to hematopoietic progenitor, both immunophenotypically (Figure 4D) and functionally, at expected levels (Figure 4B left column).

Figure 4A shows an upregulation of IGFBP2 in arterial cells as a result of TF activation. However, from the data presented here, it is not possible to evaluate whether this is specific to the arterial cluster, or it is a common effect shared by all cell types regardless of their identity.

Data has now been included in Supplementary Figure 4H, which shows that all the cells show an increase in IGFBP2, but arterial cells show the highest increase. We have now edited the text to reflect this, in lines 228-230.

In Figure 5A-B only a minority of arterial cells express RUNX1 in response to IGFBP2 treatment. Is this sufficient to explain the very significant increase in the generation of functional hematopoietic progenitors described in Figure 4? Quantification and statistical analysis of RUNX1 upregulation would strengthen this conclusion.

We have now provided the statistical analysis showing significant upregulation of RUNX1 upon IGFBP2 addition. The p values are now provided in the figure 5 legend.

In Figure 5 the authors conclude that IGFBP2 remodels the metabolic profile of endothelial cells. However, it is not clear which cell types and clusters were included in the analysis of Figure 5C-G. Is the switch from Glycolysis to Oxidative Phosphorylation specific to endothelial cells? Or it is a more general effect on the entire culture, including hematopoietic cells?

We based this conclusion on the fact that the single-cell RNAseq allows to verify that the metabolic differences are obtained in the endothelial cells. Given that we sorted the adherent cells, the majority of these are endothelial cells as shown in Figure 5A. The Seahorse pipeline includes a number of washing steps resulting in the analyses being performed on the adherent compartment which we know consists primarily of endothelial cells. We cannot exclude some contamination from non-endothelial cells but we highlight to this reviewer that the initial observation of the metabolic changes was identified in endothelial cells in the single cell sequencing data. Taken together, we believe that this implies that metabolic changes are specific to this population. We have clarified this in the line 317.

In the discussion, the authors conclude that they "provide evidence to support the hypothesis that RUNX1T1 could regulate IGFBP2 expression". To further support this conclusion, the authors could provide a correlation analysis of the expression of the two genes in the cell type of interest.

Following the observation of the IGFBP2 high expression across clusters, we have now reworded this sentence in lines 382-385 We have tried to perform the correlation analysis but we believe this not to be appropriate due to the detection level of the gRNA, we have now included this as a limitation point in the discussion lines 416-427, and also toned down the conclusion we did draw about RUNX1T1 throughout the whole manuscript.

As mentioned by the authors, IGFBP2 binds IGF1 and IGF2 modulating their function. Both IGF1 (http://dx.doi.org/10.1016/j.ymeth.2015.10.001) and IGF2 (doi:10.1038/nbt.3702) have been used in iPSC differentiation into definitive hematopoietic cells. It would be relevant to discuss/reference this in the discussion.

We have now included the suggested reference in the section where we discuss the role of IGFBP2 in binding IGF1 and IGF2.

**Reviewer #2 (Recommendations For The Authors):**
(1) Figure 1 compares the transcriptome of human AGM and in-vitro derived hemogenic endothelial cells (HECs). It is not clear why only the genes downregulated in the latter were chosen. Are there any significantly upregulated genes, knockdown/knockout which could also serve a similar purpose? Single-cell transcriptome database analysis is very preliminary. A detailed panel with differences in cluster properties of HECs between the two systems should be provided. A heatmap of all differentially expressed genes between the two samples must be generated, along with a logical explanation for choosing the given set of genes.

We have now included another panel in figure 1 to better clarify the logic behind the strategy used to identify our target genes (Figure 1A).

(2) Figure 2 - a panel describing the workflow of gRNA design and targeting for the 9 candidate genes, along with lentiviral packaging and transduction would make it easier to follow.

We have now included three schematics for all the experiments in the manuscript in supplementary figure 4 A-C.

(3) Figure 3- to assess the effect of arterial cell expansion on the emergence of hematopoietic progenitors, CD34+ Dll4+ cells should be sorted for OP9 co-culture assay.Using only CD34+ cells does not answer the question raised. Also, the CFU assay performed does not fully support the claim of enhanced hematopoietic differentiation since only CFU-E and CFU-GM colonies are increased in Dox-treated samples, with no effect on other colony types. OP9 co-culture assay with these cells would be required to strengthen this claim.

We wanted to clarify that the effect on the methylcellulose coming from the activated cells was not limited to CFU-E, as the reviewer reported; instead, it also affected CFU-GM and CFU-M.

We have now performed additional experiments where we sorted the CD34+ compartment into DLL4- and DLL4+ in Supplementary Figure 5D-E, which we discussed in lines 250-258.

(4) In Figure 3F, there appears to be a lot of variation in the DLL4% fold change values forDOX treated iSAM_AGM sample, which weakens the claim of increased arterial expansion.Can the authors explain the probable reason? It is suggested that the two other controls (iSAM_+DOX and iSAM_-DOX) should be included in this analysis. It is imperative to also show % populations rather than just fold change to gain confidence.

We agree that there is a lot of variability. That is because differentiation happens in 3D in embryoid bodies, which contain many different cell types that differentiate in different proportions across independent experiments. We have now included the raw data in Supplementary Figure 4 D, with additional statistical analysis to show the expansion of arterial cells including also the suggested additional controls.

(5) How does activation of these target genes cause increased arterialization? Is the emergence of non-HE populations suppressed? Or is it specific to the HE? The data on this should be clarified and also discussed. ANTO/Lesley text

We have provided additional data clarifying the connection between increased arterialisation and hemogenic potential. We showed that the activation induces increased arterialisation and that IGFBP2 acts by supporting the acquisition of hemogenic potential. We have discussed this in lines 326-348 and provided a new figure to explain this in detail (figure 6)

(6) Considering that IGFBP2 was chosen from the activated target gene(s) cluster, can the authors explain why the reduced CFU-M phenomenon observed in Figure 3G does not appear in the MethoCult assay for IGFBP2 treated cells (Figure 4B)?

The difference could be explained by the fact that in Figure 3G, the cells underwent activation of multiple genes, while in Figure 4B, they were only exposed to IGFBP2. Our results show that IGFBP2 could at least partially explain the phenotype that we see with the activation, but we believe that during the activation experiments, there might be other signals available that might not be induced by IGFBP2 alone. We have also added a summary section and a figure to clarify the different mechanisms of action of the gene activation and IGFBP2.

(7) Figure 4- while the experiments conducted support the role of IGFBP2 in increasing hematopoietic output, there is no experimental evidence to prove its function through paracrine signalling in HECs. The authors need to provide some evidence of how IGFBP2 supplementation specifically expands only the hematopoietic progenitors. Experimental strategies involving specifically targeting IGFBP2 in hemogenic/arterial endothelial cells are required to prove its cell type specific function. Additionally, assessing the in vivo functional potential of the hematopoietic cells generated in the presence of IGFBP2, by bone-marrow transplantation of CD34+ CD43+ cells, is essential.

The role of IGFBP2 in the context of HSC production and expansion was not the topic of our research, and we have not claimed that IGFBP2 affects the long-term repopulating capacity of HSPCs. Therefore, we believe that the requested experiments are not required to support the specific claims that we do make. We have now provided more experiments and bioinformatic analysis that support the role of IGFBP2 in inducing the progression of EHT from arterial cells to hemogenic endothelium, and to avoid misunderstandings, we have toned down our claims by editing the text regarding its paracrine effect s.

(8) Figure 4C-D -It is recommended to plot % populations along with fold change value. As this is a key finding, it is important to perform flow cytometry for additional hematopoietic markers- CD144, CD235a and CD41a to demonstrate whether this strategy can also expand erythroid-megakaryocyte progenitors. Telma

Figure 4C already shows the percentage values; we have now added the percentage for Figure 4D in SF5C. We have also performed additional analysis as requested and added the data obtained to Supplementary Figure 5D.

(9) In Figure 5, analysis showing the frequency of cells constituting different clusters, between untreated and IGFBP2-treated samples in the single-cell transcriptome analysis is essential. Additional experiments are required to validate the function of IGFBP2 through modulation of metabolic activity. Inhibition of oxidative phosphorylation in the IGFBP2treated cells should reduce the hematopoietic output. Authors should consider doing these experiments to provide a stronger mechanistic insight into IGFBP2-mediated regulation of hematopoietic emergence.

We have now included the requested cluster composition in Supplementary Figure 5F. We decided not to include further tests on the metabolic profile of IGFBP2 as we already discussed in other papers that showed, using selective inhibitors, that the EHT coincides with a glycol to OxPhos switch.

(10) It is very striking to see that IGFBP2 supplementation changes the transcriptional profile of developing hematopoietic cells by increasing transcription of OXPHOS-related genes with concomitant reduction of glycolytic signatures, particularly at Day 13. However, the mitochondrial ATP rate measurements do not seem convincing. The bioenergetic profiles show that when mitochondrial inhibitors are added, both groups exhibit decreased OCR values and, on the other hand, higher ECAR. This indicates that both groups have the capability to utilize OXPHOS or glycolysis and may only differ in their basal respiration rates.Differences in proliferation rate can cause basal respiration to change. There is no information on how the bioenergetic profile was normalized (cell no./protein amount). Given that IGFBP2 has been shown to increase proliferation, it is very likely that the cells treated with IGFBP2 proliferated faster and therefore have higher OCR. The data needs to be normalized appropriately to negate this possibility.

We have previously tested whether IGFBP2 causes an increase in proliferation by analysing the cell cycle of cells treated with it, as we initially thought this could be a mechanism of action. We have now provided the quantification of the cell cycle in the cells treated with IGFBP2, showing no effect was observed in cell cycle Supplementary Figure 4E. Following this analysis, we decided to plate the same number of cells and test their density under the microscope before running the experiment; each experiment was done in triplicate for each condition. We have now added this info to the method sections lines 806-813. We did not comment on the basal difference, which we agree might be due to several factors, but we only compared the difference in response to the inhibitors, which isn’t affected by the basal level but exclusively by their D values. We have also included the formulas used to calculate the ATP production rate.

Overall, it appears that IGFBP2 does not seem to primarily cause metabolic changes, but simply accelerates the metabolic dependency on OXPHOS. Hence, the term 'metabolic remodelling' must be avoided unless IGFBP2 depletion/loss of function analysis is shown.

We thank the reviewer for suggesting how to interpret the data about the dependency on OXPHOS. We have now changed the conclusions and claims about the effect of IGFBP2. We have also included a cell cycle analysis of the hematopoietic cells derived upon IGFBP2 addition to show that they don’t show differences in proliferation that could cause the increase in colony formation we observed. Regarding the assay, we have plated the same number of cells for each group to make sure we were comparing the same number of cells, which we also assessed in the microscope before the test, and we eliminated the suspension cells during the washes that preceded the measurement. The review is correct in indicating that there is a basal difference in the value of OCR and ECAR where the IGFBP2 is lower at the start and not higher, which would not conceal higher proliferation. Finally, the ATP production rate is calculated on the variation of OCR and ECAR upon the addition of inhibitors, which normalizes for the basal differences.